# Fine-Grained Visual Prompting

**Lingfeng Yang**[1], **Yueze Wang**[2], **Xiang Li**[3*], **Xinlong Wang**[2], **Jian Yang**[1*]

[1]Nanjing University of Science and Technology
[2]Beijing Academy of Artificial Intelligence, [3]Nankai University

{yanglfnjust, csjyang}@njust.edu.cn, {yzwang, wangxinlong}@baai.ac.cn
xiang.li.implus@nankai.edu.cn

## Abstract

Vision-Language Models (VLMs), such as CLIP, have demonstrated impressive zero-shot transfer capabilities in image-level visual perception. However, these models have shown limited performance in instance-level tasks that demand precise localization and recognition. Previous works have suggested that incorporating visual prompts, such as colorful boxes or circles, can improve the ability of models to recognize objects of interest. Nonetheless, compared to language prompting, visual prompting designs are rarely explored. Existing approaches, which employ coarse visual cues such as colorful boxes or circles, often result in sub-optimal performance due to the inclusion of irrelevant and noisy pixels. In this paper, we carefully study the visual prompting designs by exploring more fine-grained markings, such as segmentation masks and their variations. In addition, we introduce a new zero-shot framework that leverages pixel-level annotations acquired from a generalist segmentation model for fine-grained visual prompting. Consequently, our investigation reveals that a straightforward application of blur outside the target mask, referred to as the Blur Reverse Mask, exhibits exceptional effectiveness. This proposed prompting strategy leverages the precise mask annotations to reduce focus on weakly related regions while retaining spatial coherence between the target and the surrounding background. Our **F**ine-**G**rained **V**isual **P**rompting (**FGVP**) demonstrates superior performance in zero-shot comprehension of referring expressions on the RefCOCO, RefCOCO+, and RefCOCOg benchmarks. It outperforms prior methods by an average margin of 3.0% to 4.6%, with a maximum improvement of 12.5% on the RefCOCO+ testA subset. The part detection experiments conducted on the PACO dataset further validate the preponderance of FGVP over existing visual prompting techniques. Code is available at https://github.com/ylingfeng/FGVP.

## 1 Introduction

The usage of Vision-Language models (VLMs) [42, 31, 1, 30] has become increasingly prominent in various vision-related tasks, largely due to their notable impact. These models benefit from a vast amount of training data and parameters, demonstrating powerful performance in the training of fundamental visual model backbones [34, 15, 51]. Moreover, the transfer learning potential of VLMs has been employed in tasks such as open vocabulary detection [20, 64], visual grounding [32, 37], and image editing [6, 5], etc.

In the above-mentioned works, the VLMs generally require additional training, to adapt to each specific downstream task. However, the utilization of off-the-shelf Vision-Language Models (VLMs)

---

*Corresponding authors.

37th Conference on Neural Information Processing Systems (NeurIPS 2023).

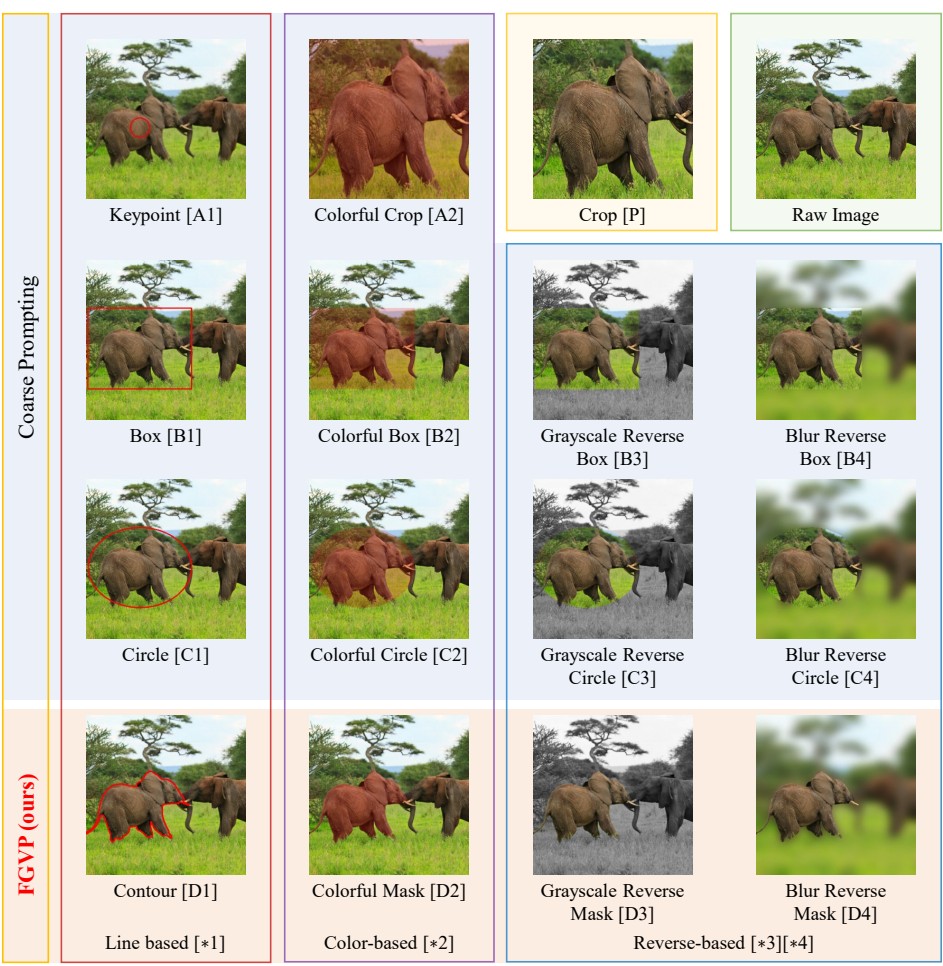

Figure 1: A Summary of visual prompts with the caption "elephant on the left".

for leveraging their inherent image-text understanding, without additional tuning, remains largely unexplored. A notable challenge lies in the low sensitivity of VLMs to perform object spatial localization due to the presence of a large amount of weakly related background noise in the images [68, 3, 35]. The common practice involves cropping the region of interest [10, 62] to obtain a zoomed-in visual representation of a single object, at the expense of discarding valuable global information. As a result, existing training-free approaches for classification lack a comprehensive understanding of contextual information and spatial awareness across different objects.

In recent studies, CPT [61] and ReCLIP [50] have leveraged visual prompting techniques to establish relations between partial instances. Visual prompting is a technique used in image-language tasks, where markers such as colorful boxes or circles are added directly onto an image to highlight specific targets. By utilizing appropriate visual prompts, the attention of VLMs can be effectively directed toward the desired targets while preserving the global context. Moreover, a study by RedCircle [49] demonstrated that drawing red circles enclosing the object on the entire image can effectively distinguish instances, where the circles correspond to inscribed ellipses derived from proposal boxes. This discovery suggests that Vision-Language Models (VLMs) may possess the inherent ability to understand the local object within an overall image. Therefore, a specifically designed visual prompt has the potential to explicitly invoke this capability of VLMs, thereby benefiting various tasks.

Despite the interest in visual prompting techniques, their unique designs have yet to be fully explored. The current approaches only rely on coarse markers like colorful boxes or circles, which can introduce ambiguity and fail to highlight the intended instance accurately. In this paper, we address this issue by systematically organizing and investigating various forms of visual prompting. Further, we propose a refined technique called Fine-Grained Visual Prompting (FGVP), which utilizes semantic masks to mark each target precisely, thereby enhancing zero-shot performance.

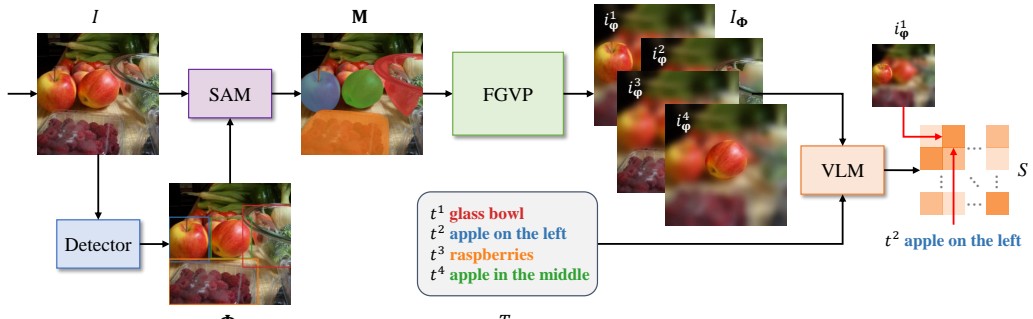

Figure 2: Structure of fine-grained visual prompting with box proposals from a detector.

In our research, we examine different visual prompting designs showcased in Fig. 1, including those utilized in previous studies such as CPT [61], ReCLIP [50], and RedCircle [40]. To provide an overview of previous research, we summarize their strategies in Table 1. It is important to note that some approaches achieve their best results through prompt ensembles or post processing techniques, which are discussed extensively in Sec. 3.1. Despite the progress made in visual prompting, none of the existing methods have fully explored the use of more precise semantic masks as fine-grained markers, rather than relying on coarse boxes or circles. To verify the effectiveness, we begin by exploring the upper bound of these prompts. Our methodology assumes perfect accuracy of all annotations applied, ensuring that images are marked with ground-truth information, namely bounding boxes and masks. We confirm that a *single* Blur Reverse Mask [$D4$], which involves blurring all content outside the instance mask, significantly outperforms other types of visual prompts on multiple datasets. This indicates the effectiveness of our fine-grained visual prompt design.

In practical scenarios where ground-truth annotations are unavailable, there are two primary approaches for obtaining fine-grained visual markings. The first approach is based on utilizing object proposals predicted by a pretrained detector, following a similar paradigm as employed in MAttNet [62]. This approach is commonly employed in zero-shot classification tasks [61, 50, 49]. To generate fine-grained masks, we employ Segment Anything Model (SAM) [26], which takes the aforementioned bounding boxes (refer to Fig. 2) as input prompts. Additionally, we propose a second approach that eliminates the dependency on a detector for prompt candidates. Our approach establishes a viable zero-shot pipeline by solely leveraging SAM, irrespective of specific tasks. Specifically, we prompt SAM with grid-wise keypoints and subsequently apply a non-maximum suppression (NMS) operation to obtain proposals with semantic masks. These fine-grained masks can then be employed for zero-shot classification (see Fig. 3).

| Existing Method | Visual Prompt | Post Processing |
|---|---|---|
| MAttNet [62], UNITER [10] | $P$ | – |
| CPT [61] | $A2$ | – |
| ReCLIP [50] | $P \mid B4$ | Relations [50] |
| RedCircle [40] | $C1 \mid C3 \mid C4$ | Subtraction [40] |

Table 1: The type of Visual Prompt equipped in previous works related to the referring expression comprehension task. "$\mid$" denotes ensemble results of multiple prompts.

Our contributions can be summarized as follows:

• We propose Fine-Grained Visual Prompting (FGVP), employing Blur Reverse Masks to enhance the semantic localization capability of Vision-Language models such as CLIP (Fig. 1). To the best of our knowledge, we are the first to explore the use of Blur Reverse Masks as visual prompting, highlighting their remarkable effectiveness in zero-shot image-text comprehension.

• We are the first to provide a comprehensive exploration of visual prompt formats, including crop, box, circle, mask with different line markings, colored masks, grayscale, and Gaussian blurring. Additionally, we thoroughly analyze and evaluate the impact of associated auxiliary attributes such as blur deviation, color, etc., ensuring a comprehensive understanding of their effects.

• Our proposed Fine-Grained Visual Prompting (FGVP) achieves state-of-the-art (SOTA) zero-shot results on the referring expression comprehension task, surpassing ReCLIP and RedCircle by an average of 4.6% and 3.0% accuracy, respectively, across RefCOCO, RefCOCO+, and RefCOCOg benchmarks. Notably, our zero-shot pipeline demonstrates superior part detection accuracy on the PACO dataset compared to previous visual prompting methods.

## 2 Related Work

**Vision-Language Models**   Large Language Models (LLMs), such as GPT-3 [7], GPT-4 [41], LLaMA [53], and PaLM [12], have demonstrated strong zero-shot transfer abilities in natural language processing. In recent years, vision-language Models (VLMs) that leverage image-text data pairs from the web have gained prominence in computer vision (CV) tasks. CLIP [42] and ALIGN [31] learn image-text alignment through contrastive learning. Furthermore, models like Flamingo [1] have shown impressive few-shot learning capabilities. BLIP-2 [30, 29] proposes joint multimodal task handling through transfer learning. Notably, VLMs have excelled in image-text generation as demonstrated by DALL-E [46, 45]. However, instance-level tasks such as referring expression comprehension and part detection typically require tuning of vision and text encoders in existing open vocabulary methods [20, 64] and image grounding approaches [32, 37]. In contrast, this paper proposes a zero-shot architecture for instance-level tasks using off-the-shelf VLMs.

**Prompt Engineering**   Prompt engineering is a well-developed practice in NLP [43, 7]. Auto-Prompt [48] and CoOp [69] automate the generation of prompt templates, leveraging gradient-based methods instead of manual crafting. Language prompting is then extended to open vocabulary detection [14, 16] and segmentation [59]. While language prompting has been extensively explored, visual prompting has received less attention. Previous works [35, 22, 2, 4] use visual prompt tuning to adapt to VLMs. In terms of zero-shot solutions, CPT [61] introduces colorful boxes as markers on images, and RedCircle [49] demonstrates the effectiveness of a circle mark for visual attention during zero-shot classification. However, existing visual prompting methods lack fine-grained precision. In contrast, we propose leveraging semantic masks derived from segment models like Segment Anything [26] for more precise visual marking. It is worth noting that CPT [61] also employs semantic masks as colored prompts, but only after cropping the region using bounding boxes. In our approach, we directly mark the fine-grained mask on the entire image to preserve the global visual content.

**Image Segmentation**   Image segmentation is a common task in computer vision [52], involving predicting region segment masks by labeling pixels. This task encompasses various sub-tasks such as instance segmentation [21, 13, 55], semantic segmentation [38, 67, 11], and panoptic segmentation [25, 25]. Recent advancements include SegGPT [56], which facilitates performing diverse segmentation tasks through in-context visual learning. Another notable approach, Segment Anything Model (SAM) [26], introduces spatial prompts for segmenting arbitrary objects. SAM is trained using an extensive dataset of over 1 billion segmentation masks. In our study, we employ the off-the-shelf SAM framework to generate precise semantic masks for fine-grained visual prompting.

**Referring Expression Comprehension**   This task involves providing captions for specific objects in images and localizing them with bounding boxes. Proposal-based methods use object detectors like Mask R-CNN [21] to detect instance regions. Each proposal is then cropped and used for subsequent categorization. Examples include MAttNet [62], UNITER [10], OSCAR [33], and VinVL [66]. Proposal-free methods, such as MDETR [23] and ViLT [24], train Vision-Language Models end-to-end. Recent zero-shot methods [61, 50, 40] combine proposal boxes from MAttNet [62] with Vision-Language Models for image-caption matching. Our work follows this setup for fair comparisons.

**Part Detection**   Part Detection is a sub-field of object detection [21, 47, 17, 8], primarily focused on fine-grained classification [65, 54] and human parsing [58, 18] that require precise region localization. Notable datasets include CUB-200 [54] for annotated bird parts, LIP [19] for semantic human part labels, PASCAL-Part [9] for common objects, and PACO [44], which introduces a comprehensive benchmark with jointly annotated part masks. Previous zero-shot works typically detect objects and parts using pre-annotated visual attributes to transfer the knowledge [28]. In contrast, we leverage the transferability of Vision-Language Models to perform part detection in this study.

## 3 Method

In this section, we begin by presenting a comprehensive overview of the visual prompting pipeline for zero-shot object recognition and part detection tasks. Subsequently, we delve into the details of our Fine-Grained Visual Prompting (FGVP) architecture. Finally, we examine the effectiveness and rationale behind our design.

## 3.1 Framework

The zero-shot framework takes as input an image $I \in \mathbb{R}^{3 \times H \times W}$, $N$ box proposals $\boldsymbol{\Phi} \in \mathbb{R}^{N \times 4}$ and $M$ caption texts $T \in \Omega^M = \{t^1, t^2, ..., t^M\}$, where $\Omega$ denotes the set of textual sentences. The goal is to find the best matching image-text pairs. A common practice is to get each image input by cropping or RoIAlign [21] according to $\boldsymbol{\Phi}$. However, with the introduction of visual prompting, one can mark the regions on the whole image, which highlights each target instance while keeping the background knowledge. For simplicity, we regard cropping as an ordinary type of visual prompting. Then the image input $I_{\boldsymbol{\Phi}} \in \mathbb{R}^{N \times 3 \times H \times W} = \{i_{\boldsymbol{\varphi}}^1, i_{\boldsymbol{\varphi}}^2, ..., i_{\boldsymbol{\varphi}}^N\}$ for VLMs can be generated as:

$$I_{\boldsymbol{\Phi}} = \mathbf{VP}(I, \boldsymbol{\Phi}), \tag{1}$$

where $\mathbf{VP}$ concludes visual prompting, such as cropping, drawing colorful boxes or circles, or employing fine-grained masks, etc. Then the cosine similarity $S = \{s(i_{\boldsymbol{\varphi}}^n, t^m)\}_{N \times M}$ between each image proposal $I_{\boldsymbol{\Phi}}$ and text $T$ can be derived by the VLM, e.g., CLIP [42]. For the referring expression comprehension task, $T$ is a gathering of short sentences that describe the related instances in the image, and one needs to predict their referring location. Note that there exist some post processing techniques for this task, namely "Relations" [50] and "Subtraction" [49], respectively. The "Relations" considers spatial relations $R = \{r(i_{\boldsymbol{\varphi}}^{n_1}, i_{\boldsymbol{\varphi}}^{n_2})\}_{N \times N}$ between every two proposals, w.r.t. left, right, above, below, bigger, smaller, and inside. Then the parsed final scores are updated as:

$$S = \{\boldsymbol{\Sigma}_{i_{\boldsymbol{\varphi}}^{n'} \in I_{\boldsymbol{\Phi}}}(s(i_{\boldsymbol{\varphi}}^n, t^m) \cdot r(i_{\boldsymbol{\varphi}}^n, i_{\boldsymbol{\varphi}}^{n'})), s \in S, r \in R\}. \tag{2}$$

Besides, the "Subtraction" deals with positive and negative text set to weigh down the score. By randomly sampling Q negative captions $\widetilde{T}$ that related to no instances on the image, all negative scores are calculated as $\widetilde{S} \in \mathbb{R}^{N \times Q} = \widetilde{s}(i_{\boldsymbol{\varphi}}^n, \widetilde{t}^q)$. And then use it to penalize $S$:

$$S = \{s(i_{\boldsymbol{\varphi}}^n, t^m) - \frac{1}{Q} \cdot \boldsymbol{\Sigma}_{\widetilde{t}^q \in \widetilde{T}} \widetilde{s}(i_{\boldsymbol{\varphi}}^n, \widetilde{t}^q), s \in S, \widetilde{s} \in \widetilde{S}\}. \tag{3}$$

Once the final scores are obtained, the referring region $\widehat{\boldsymbol{\Phi}}$ can be given by:

$$\widehat{\boldsymbol{\Phi}} = \{\widehat{\varphi} \mid \widehat{\varphi} = \mathbf{ARGMAX}_{i_{\boldsymbol{\varphi}}^n \in I_{\boldsymbol{\Phi}}} s(i_{\boldsymbol{\varphi}}^n, t^m), s \in S\}. \tag{4}$$

For the part detection task, the text is the object or part label names. Considering the general case, each part only appears in one region (all eyes of a bird are covered by a single box). Thus we indicate a post processing strategy for this task with the Hungarian algorithm [27], to find a bipartite matching between image proposals and part labels. Otherwise, the label predicted corresponding to each proposal is derived as:

$$\widehat{T} = \{\widehat{t} \mid \widehat{t} = \mathbf{ARGMAX}_{t^m \in T} s(i_{\boldsymbol{\varphi}}^n, t^m), s \in S\}. \tag{5}$$

## 3.2 Fine-Grained Visual Prompting

Existing visual prompts primarily use proposal boxes from the detector. The Crop $[P]$ and Colorful Crop $[A2]$ prompts involve extracting cutouts of the original image based on the location of the box. Prompting with Keypoint $[A1]$ entails placing a small circle around the box center. The box-based $[B*]$ prompts naturally use the box as the marking boundary, while the circle-based $[C*]$ prompts are essentially similar to $[B*]$, except for replacing the box with its inscribed ellipse.

However, these prompts are too coarse to emphasize key targets. Marking inaccuracies introduce irrelevant information and lead to sub-optimal performance based on empirical evidence. We illustrate previous visual prompting approaches as coarse prompting on top of Fig. 1. In contrast, we investigate fine-grained visual prompting using semantic masks, which can accurately highlight the target instance, reduce background interference, and retain global knowledge. Nonetheless, obtaining semantic masks may pose a challenge when only proposal boxes are provided. To tackle this issue, we propose the use of a robust image segmentation model called Segment Anything Model (SAM) [26] to perform class-agnostic segmentation based on the given boxes (Fig. 2):

$$\mathbf{M} = \mathbf{SAM}(I, \boldsymbol{\Phi}), \tag{6}$$
$$I_{\boldsymbol{\Phi}} = \mathbf{FGVP}(I, \mathbf{M}), \tag{7}$$

where $\mathbf{SAM}$ denotes Segment Anything Model [26], and $\mathbf{M} \in \mathbb{R}^{N \times H \times W}$ denotes the semantic masks. We updated Eq. (1) with Eq. (6), (7) to produce fine-grained visual prompting.

In addition, We innovatively proposed a zero-shot classification pipeline, which does not require pre-processed box proposals but directly generates fine-grained markers. The key idea lies in that

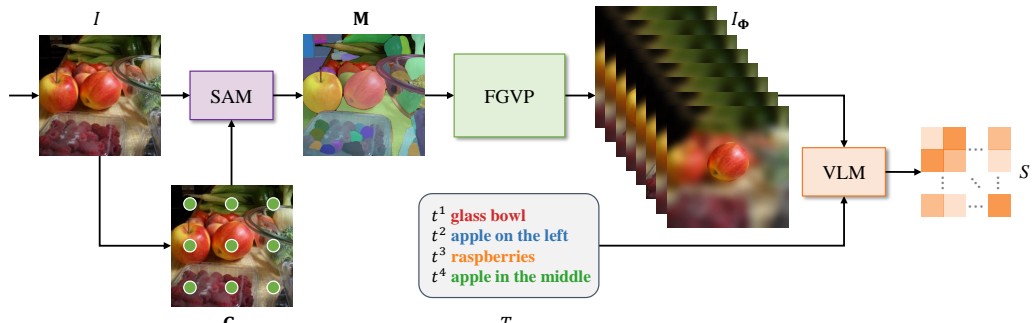

Figure 3: Structure of fine-grained visual prompting with no box proposal. Masks are directly derived via SAM prompted by grid-wise keypoints.

SAM can propose an extremely exhaustive prediction of almost any object of parts on the images taken a grid-wise set of keypoints $\mathbf{G} \in \mathbb{R}^{g^2 \times 2}$ as inputs (Fig. 3), where $g$ is the point number along one side of the image. Then the mask is generated through SAM prompted by the grid-wise keypoints:

$$\mathbf{M} = \mathbf{NMS}(\mathbf{SAM}(I, \mathbf{G})), \tag{8}$$

where $\mathbf{NMS}$ denotes Non-Maximum Suppression to filter out duplicate proposals. Notably, the regression box $\mathbf{\Phi}$ is obtained by calculating the smallest box containing the object mask.

Following the FGVP illustration, we provide an overview of all visual prompting variants. In addition to the categorization based on box, circle, and mask, they can also be sorted based on different markers, such as line [∗1], color [∗2], grayscale [∗3], and blur [∗4]. Specifically, line-based methods utilize closure lines to prompt the image, while color-based prompting involves drawing colorful masks on the target. These two types can be classified as positive marking, which aims to highlight the target area. Conversely, the reverse-based prompting serves as a form of negative marking by grayscaling or blurring the background area, thereby reducing the impact of weakly related information.

## 3.3 Discussion

In fact, both part detection and referring tasks require a mutual understanding between instances and backgrounds. In the referring expression comprehension task, captions are used to describe the interrelationships between multiple objects. Similarly, in part detection, a local part can be hard to classify regardless of the object. Successful performance of these tasks thus necessitates a network's ability to handle global information and accurately comprehend the relationships between instances. For this reason, implementing more precise visual prompting is helpful, typically with the Blur Reverse Mask Prompting [$D4$]. This is because the web-scale dataset on which VLM is trained contains a large amount of photography.

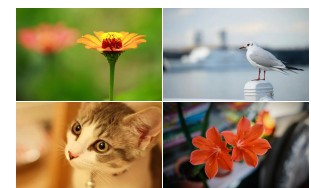

Figure 4: Photography from the Internet with "Bokeh".

These images prefer to employ "Bokeh" to blur the background and highlight the subject (Fig. 4). As a result, VLMs may have prior knowledge of visual prompting with background blurring.

## 4 Experiments

In this section, we first evaluate individual visual prompting performance. Then, we compare FGVP with previous zero-shot methods on the referring expression comprehension and part detection tasks to show our effectiveness. For more experimental analysis refer to the Supplementary Materials.

### 4.1 Dataset

We conduct the experiments on several visual datasets, i.e., RefCOCO [63], RefCOCO+ [63], RefCOCOg [39], COCO [36], and PACO [44]. The RefCOCO, RefCOCO+, and RefCOCOg datasets are subsets selected from COCO, containing bounding boxes and masks corresponding to captioned

| Visual Prompt | Ground Truth | | | | | Referring Expression Comprehension | | |
|---|---|---|---|---|---|---|---|---|
| | COCO | PACO | RefCOCO | RefCOCO+ | RefCOCOg | RefCOCO | RefCOCO+ | RefCOCOg |
| Crop | **70.9** | 38.5 | 35.2 | 40.3 | 59.1 | 45.3 | 46.4 | 56.4 |
| Keypoint | 52.3 | 39.1 | 36.9 | 39.6 | 43.8 | 46.7 | 47.9 | 48.9 |
| Colorful Crop | 64.2 | 35.7 | 37.1 | 41.9 | 58.0 | 48.2 | 49.0 | 57.0 |
| Box | 48.5 | 42.7 | 34.7 | 39.5 | 44.6 | 45.5 | 46.4 | 47.0 |
| Colorful Box | 34.4 | 37.2 | 23.9 | 23.4 | 22.7 | 35.4 | 30.7 | 30.8 |
| Grayscale Reverse Box | 42.4 | 37.4 | 34.4 | 35.9 | 44.5 | 45.9 | 44.0 | 48.4 |
| Blur Reverse Box | 62.1 | 39.2 | 47.9 | 51.8 | **63.6** | 48.8 | 51.4 | 54.1 |
| Circle | 48.9 | 42.6 | 43.2 | 49.3 | 56.3 | 48.9 | 51.7 | 54.6 |
| Colorful Circle | 36.1 | 37.2 | 29.9 | 29.8 | 24.5 | 40.7 | 37.1 | 37.9 |
| Grayscale Reverse Circle | 42.9 | 36.6 | 36.9 | 38.2 | 47.3 | 47.8 | 46.2 | 50.3 |
| Blur Reverse Circle | 58.1 | 36.8 | 49.2 | 53.1 | 60.9 | 49.3 | 52.1 | 52.2 |
| Contour | 47.3 | 41.0 | 38.7 | 41.7 | 43.5 | 46.1 | 45.0 | 46.3 |
| Mask | 41.1 | **43.7** | 29.9 | 29.1 | 29.9 | 41.8 | 38.5 | 38.4 |
| Grayscale Reverse Mask | 45.2 | 40.4 | 40.5 | 43.8 | 50.9 | 45.8 | 45.9 | 51.2 |
| Blur Reverse Mask | 67.8 | 43.3 | **52.8** | **58.0** | 63.5 | **52.8** | **55.4** | **57.8** |

Table 2: Ablation study on the zero-shot performance of individual visual prompting in the validation set of COCO, PACO, RefCOCO, RefCOCO+, and RefCOCOg datasets using ground truth annotations (left) and proposals in referring expression comprehension (right), respectively. Best metrics are in **bold**, and sub-optimal results are underlined.

instances. The COCO dataset is annotated with boxes and masks for objects, while PACO additionally incorporates annotations of corresponding parts for each object.

## 4.2 Implementation Details

We follow Timm [57], CPT [61], and ReCLIP [50] to construct our pipelines. By default, we use ViT-B/32, ViT-L/14@336px, and RN50x16 backbones trained in CLIP [42] by OpenAI as our vision-language model. To simplify reference to these models in subsequent experiments, we refer to them as ViT-B, ViT-L, and RN. Additionally, for improved mask generalization, we employ SAM-ViT-H, a variant of Segment Anything Model [26]. Concerning part detection on the PACO [44] dataset, we begin by cropping the object according to the annotated boxes and then perform part detection on each object crop. All experiments are conducted on $8\times$ Tesla V100.

## 4.3 Ablation Study on Visual Prompt

In this section, we elaborate on the zero-shot performance of individual visual prompts defined in Fig. 1 with two experiment settings. Firstly, we assume all prompting representations, i.e., bounding boxes and semantic masks, to be as precise as possible, in order to explore the upper bound of each visual prompt. Therefore, we acquire them from ground truth annotations of all datasets. Then we evaluate the image-label matching accuracy following the zero-shot pipeline described in Sec. 3.1. More specifically, instance-label pairs are represented as object-object name, part-part name, and object-caption in COCO, PACO, and RefCOCO series datasets, respectively. The left part of Table 2 reveals that the Blur Reverse Mask [D4] offers the best overall performance.

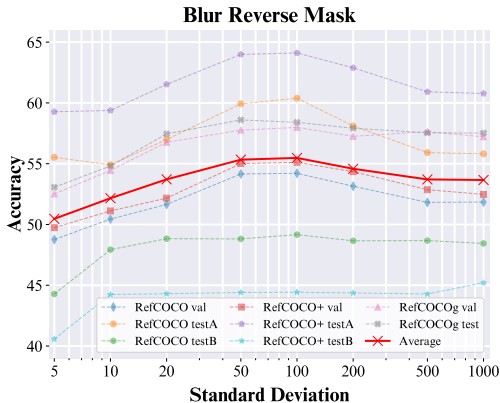

Figure 5: Ablation on the standard deviation in Gaussian blur kernel from the Blur Reverse Mask [D4] prompting. A larger deviation presents a more blurred background.

In the second part, we ablate on the referring expression comprehension task under the codebase of ReCLIP [50] with proposal boxes provided by MAttNet [62]. Note that in this setting, semantic

| Method | Backbone | Visual Prompt | PP | RefCOCO | | | RefCOCO+ | | | RefCOCOg | |
|---|---|---|---|---|---|---|---|---|---|---|---|
| | | | | val | testA | testB | val | testA | testB | val | test |
| *Our codebase with object proposals detected by UNINEXT [60]* | | | | | | | | | | | |
| Crop | ViT-L | $P$ | – | 45.3 | 47.2 | 43.7 | 46.4 | 48.7 | 42.9 | 56.4 | 56.3 |
| RedCircle [49] † | ViT-L | $C1$ | – | 48.9 | 56.4 | 41.3 | 51.7 | 59.5 | 41.8 | 54.6 | 54.7 |
| FGVP (ours) | ViT-L | $D4$ | – | **52.8** | **56.6** | **46.4** | **55.4** | **62.0** | **46.6** | **57.8** | **58.3** |
| *CPT [61] codebase with object proposals detected by MAttNet [62]* | | | | | | | | | | | |
| CPT [61] | VinVL [66] | $A2$ | – | 32.2 | 36.1 | 30.3 | 31.9 | 35.2 | 28.8 | 36.7 | 36.5 |
| RedCircle [49] † | ViT-L | $C1$ | – | 38.0 | 45.3 | 32.9 | 43.9 | 51.0 | 37.1 | 47.2 | 47.3 |
| FGVP (ours) | ViT-L | $D4$ | – | **46.1** | **53.0** | **40.4** | **50.4** | **57.5** | **42.6** | **54.5** | **54.1** |
| *ReCLIP [50] codebase with object proposals detected by MAttNet [62]* | | | | | | | | | | | |
| CPT-adapted [50] ♮ | ViT-B, RN | $B2$ | R | 23.2 | 21.4 | 27.0 | 23.9 | 21.6 | 25.9 | 22.3 | 23.7 |
| CPT-adapted [50] ♮ † | ViT-B, RN | $P \mid B2$ | R | 41.3 | 40.6 | 44.0 | 41.3 | 41.8 | 41.1 | 51.3 | 51.2 |
| ReCLIP [50] | ViT-B, RN | $P \mid B4$ | R | 45.8 | 46.1 | 47.1 | 47.9 | 50.1 | 45.1 | 59.3 | 59.0 |
| RedCircle [49] † | ViT-B, RN | $P \mid C1$ | R | 43.9 | 46.2 | 44.1 | 45.3 | 47.9 | 43.1 | 57.3 | 56.3 |
| FGVP (ours) | ViT-B, RN | $P \mid D4$ | R | 52.0 | 55.9 | 48.8 | 53.3 | 60.4 | 46.7 | 62.1 | 61.9 |
| RedCircle [49] | ViT-L, RN | $C1 \mid C3 \mid C4$ | S | 49.8 | 58.6 | 39.9 | 55.3 | 63.9 | 45.4 | 59.4 | 58.9 |
| RedCircle [49] † | ViT-L, RN | $C1 \mid C3 \mid C4$ | S | 51.4 | 58.3 | 40.9 | 56.3 | 63.6 | 45.8 | 58.3 | 58.0 |
| FGVP (ours) | ViT-L, RN | $D1 \mid D3 \mid D4$ | S | 52.9 | 59.6 | 43.9 | 57.4 | 64.8 | 46.3 | 58.1 | 58.3 |
| RedCircle [49] † | ViT-L, RN | $P \mid C1 \mid C3 \mid C4$ | S | 51.6 | 58.0 | 42.0 | 58.1 | 64.5 | 47.5 | 60.0 | 59.3 |
| FGVP (ours) | ViT-L, RN | $P \mid D1 \mid D3 \mid D4$ | S | 53.9 | 60.2 | 44.3 | 59.3 | 66.6 | 48.8 | 61.0 | 61.3 |
| RedCircle [49] † | ViT-L, RN | $P \mid C1 \mid C3 \mid C4$ | RS | 56.8 | 62.4 | 49.1 | 58.6 | 64.7 | 48.3 | 62.2 | 61.0 |
| FGVP (ours) | ViT-L, RN | $P \mid D1 \mid D3 \mid D4$ | RS | **59.6** | **65.0** | **52.0** | **60.0** | **66.8** | **49.7** | **63.3** | **63.4** |

Table 3: The performance of the *rec* benchmarked with RefCOCO, RefCOCO+, and RefCOCOg datasets. The Visual Prompting strategies are shown in Fig. 1. **PP**: Post Processing, "R" and "S" denote Spatial Relations [50] and Score Subtraction [49], respectively. **FGVP**: Fine-Grained Visual Prompting. ♮CPT-adapted is an adapted version of CPT-Blk implemented by ReCLIP. † Re-implementation performance. The best result for each dataset, w.r.t. each codebase is in **bold**.

masks in FGVP are derived following our proposed framework presented in Fig. 2. Consequently, the Blur Reverse Mask [$D4$] shows a consistent superiority demonstrated in the right part of Table 2. Notably, in line-based [∗1] prompting, we employ a default red line with a thickness of 2 pixels, as described in the RedCircle [49]. As for color-based methods [∗2], we utilize a green color with a transparency level of 0.5 following CPT [61]. Next, we ablate the standard deviation of the Gaussian blur kernel for blur-based prompting [∗4] (Fig. 5), and a value of 100 achieves the best. More ablation study details on the prompting properties are provided in the Supplementary Materials.

## 4.4 Referring Expression Comprehension

In this section, we compare our FGVP with previous zero-shot methods in Table 3. For fair comparisons, we also implement FGVP upon the original codebases of each compared method, namely CPT [61] and ReCLIP [50].

We conduct experiments following consistent settings with each compared work. To be specific, with the CPT and our codebase, we focus on individual visual prompting performance without post-processing. We use proposals from UNINEXT and MAttNet to demonstrate the robustness of our enhancements. It's important to note that different proposal selections solely affect the box candidates, which are equitably shared among all the comparison prompting methods. Next, with the ReClip codebase, ReClip employs cropping and colorful boxes as visual prompts, with default spatially-relations post-processing. To ensure a fair comparison, we first add cropping as an ensemble to all experiments. Next, to facilitate comparison with RedCircle (which inherently uses Score Subtraction as post-processing and primarily ensembles based on three circle prompts), we adopt the same three types of prompt formats but based on semantic masks. Finally, we explore higher performance possibilities. The preprocessing procedure for text inputs, including the prefix and clear-text principle, remains consistent across all codebases. Additionally, we compare zero-shot

| Visual Prompt | PACO | RefCOCO | RefCOCO+ | RefCOCOg |
|---|---|---|---|---|
| Crop | 16.5 | 17.7 | 21.6 | 34.3 |
| Keypoint | 11.9 | 16.7 | 18.5 | 19.7 |
| Circle | 17.4 | 24.9 | 29.8 | 32.4 |
| Colorful Mask | 15.2 | 24.1 | 21.4 | 18.6 |
| Blur Reverse Mask | **18.3** | **40.8** | **44.9** | **49.6** |

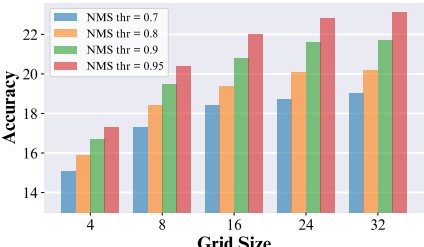

Table 4: Accuracy of the part detection with ViT-L on the validation set of each benchmark. The best result is in **bold**.

Figure 6: Ablation study on the NMS threshold and grid size.

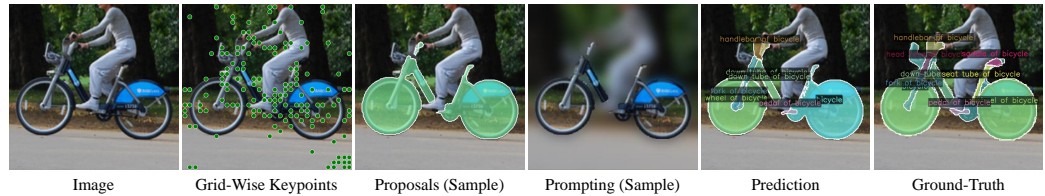

| Image | Grid-Wise Keypoints | Proposals (Sample) | Prompting (Sample) | Prediction | Ground-Truth |

Figure 7: Visualization of the candidate grid-wise keypoints, the proposals, the visual prompting image, and the predicted results in part detection.

referring accuracy under unified CLIP models. Unless otherwise stated, all existing method results are reported according to their original papers.

Table 3 shows that the single fine-grained visual prompt, i.e., Blur Reverse Mask can surpass previous works. With a ViT-L model, our FGVP surpasses RedCircle by an average of 3.4% accuracy in our codebase and 5.5%~8.1% with the CPT [61] codebase. Further, we implement FGVP under the ReCLIP [50] codebase, which investigates the ensemble of multiple models, prompts, and post processing. The results show that the improvement brought by FGVP is orthogonal to the ensemble techniques. Under equal comparisons, our FGVP shows better performance than previous works with a consistent gain of an average of ~3% and a max of 12.5% in the RefCOCO+ testA set.

### 4.5 Part Detection

The PACO [44] dataset features annotations for boxes and masks for common objects and their corresponding parts. For the part detection task, models need to locate the part within its object. Different from the referring task, there is no prior information indicating where the parts are. Our proposed pipeline (Fig. 3) is capable of performing zero-shot part detection without box proposals and utilizing only image inputs in these circumstances. Different from the localization keypoints operation employed in RedCircle [40] which only predicts the center location, we instead predict precise semantic masks of the target. Same with the metric in referring expression comprehension, a predicted box is considered correct only when the intersection over union with the ground truth box exceeds 0.5. As shown in Table 4, our proposed method, FGVP, outperforms other coarse prompts for part detection. Notably, we set the grid size to 16 along one side of the image and used an NMS threshold of 0.7 by default. However, better performance can be achieved by including more proposals through the use of a larger grid size and NMS threshold. We conduct an ablation study on them by varying one while fixing the other, as illustrated in Fig. 6. A visualization of the FGVP results is depicted in Fig. 7.

In addition, the referring expression comprehension task can be seen as part detection (each referring instance can be seen as a part of the image) when no prior box proposals are provided. We conducted zero-shot experiments on the validation set of RefCOCO, RefCOCO+, and RefCOCOg datasets, utilizing only images as inputs. As shown in Table 4, the Blur Reverse Mask Prompting without prior box proposals even surpasses certain coarse visual prompting methods with box proposals.

### 4.6 Limitations

Firstly, FGVP takes longer for inference than other methods because it involves the segmentor to produce semantic masks. To be specific, we compare the inference costs in terms of computation and

| Method | SAM scale | Mask-filter | CUDA Memory | Inference Time | IPS | Acc |
|--------|-----------|-------------|-------------|----------------|-----|-----|
| Crop | – | – | 0.91 GB | 4.49 min | 5.03 | 45.3 |
| RedCircle [49] | – | – | 0.91 GB | 4.00 min | 5.64 | 48.9 |
| FGVP (ours) | base | ✗ | 1.32 GB | 5.20 min | 4.34 | 51.7 |
| | | ✓ | 1.32 GB | 27.47 min | 0.82 | 52.1 |
| | large | ✗ | 2.14 GB | 6.29 min | 3.59 | 51.0 |
| | | ✓ | 2.14 GB | 27.49 min | 0.82 | 52.2 |
| | huge | ✗ | 3.42 GB | 7.34 min | 3.08 | 51.9 |
| | | ✓ | 3.42 GB | 28.02 min | 0.81 | 52.8 |

Table 5: Comparisons of inference cost. **IPS**: Image per GPU second.

speed between our method and others. Further, we conduct an ablation study on the scalability of SAM. Notably, the post-processing technique to filter small disconnected regions and holes in masks can further improve performance at the cost of speed. Disabling the mask-filter post-processing will greatly improve the speed without losing too much performance (Table 5). Experiments are run on RefCOCO with a CLIP pre-trained ViT-L/14@336px on $8\times$ NVIDIA A100. Above all, speeding up the architecture is an important direction for future improvement.

Secondly, the current implementation does not couple the visual prompt with a specifically designed language prompt. Therefore, the image-text alignment comprehension ability of VLM has not been fully explored appropriately. Another consideration is that there are more possible tasks that can adopt the proposed method for zero-shot transfer, such as referring image segmentation. We leave it to our future work.

# 5 Conclusion

In this paper, we focus on the visual prompt, a technique to highlight the target instances in the image content using visible markings such as colorful boxes and circles. The visual prompt is beneficial in invoking potential spatial comprehension within VLMs like CLIP on instance-level tasks. However, existing prompting designs are often too coarse for locating the target instance, leading to unrelated information that may harm performance. Since the topic of visual prompting is rarely explored, we systematically summarize various typical prompting formats and propose Fine-Grained Visual Prompting (FGVP), which utilizes precise semantic masks of target instances derived via Segment Anything (SAM). We discover that the Blur Reverse Mask prompting, which blurs the background, achieves the best performance. Furthermore, we construct two zero-shot classification architectures for regular referring expression comprehension tasks using boxes as prior proposals and for part detection utilizing only input images. We evaluate the effectiveness of FGVP on referring expression comprehension and achieve state-of-the-art (SOTA) performance.

# Acknowledgement

This project is supported by the National Key R&D Program of China (2022ZD0116302), the Young Scientists Fund of the National Natural Science Foundation of China (Grant No. 62206134), the Fundamental Research Funds for the Central Universities (070-63233084), and the Tianjin Key Laboratory of Visual Computing and Intelligent Perception (VCIP). Computation is partially supported by the Supercomputing Center of Nankai University (NKSC). Note that Lingfeng Yang and Jian Yang are with Jiangsu Key Lab of Image and Video Understanding for Social Security, and Key Lab of Intelligent Perception and Systems for High-Dimensional Information of Ministry of Education, Nanjing University of Science and Technology, Nanjing, 210094, P.R. China, while Lingfeng Yang and Xiang Li are with IMPlus@PCALab.

# Broader Impacts

Further research and careful consideration are necessary when utilizing this technology, as the presented proposed method relies on statistics derived from training datasets that may possess biases and could potentially result in negative societal impacts.

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
