# Supplementary Materials for "Fine-Grained Visual Prompting"

**Lingfeng Yang[1], Yueze Wang[2], Xiang Li[3*], Xinlong Wang[2], Jian Yang[1*]**

[1]Nanjing University of Science and Technology
[2]Beijing Academy of Artificial Intelligence, [3]Nankai University

{yanglfnjust, csjyang}@njust.edu.cn, {yzwang, wangxinlong}@baai.ac.cn
xiang.li.implus@nankai.edu.cn

## A    More Ablations on Visual Prompting Properties

In this section, we provide detailed elaborations on the settings of various visual markers.

### A.1    Thickness and Color for Line-Based Prompting

The research conducted by Shtedritski et al. [4] highlights the strong performance of the visual prompt featuring a red circle in the referring expression comprehension task. To examine the line-based visual prompting settings, we perform independent ablation experiments on line thickness and color using red circle prompting as a representative. Initially, we conduct an ablation study by varying the thickness from 1 to 5 pixels, keeping the color fixed as red. Similarly, regarding the experiment of line color, the options include $((255, 0, 0), Red)$, $((0, 255, 0), Green)$, $((255, 255, 0), Yellow)$, $((0, 255, 255), Cyan)$, $((0, 0, 255), Blue)$, and $((128, 0, 128), Purple)$. It's worth noting that CPT [6] previously conducted a grid search in the RGB space and determined the best color to be a dark red with RGB values of $(240, 0, 30)$, referred to as $CPT\text{-}Red$ in our experiment. In addition, we maintain a constant thickness of 2 pixels during the color ablation test. Finally, based on the analysis in Fig. S1 and Fig. S2, we observe that a thickness of 2 pixels and a red color achieve the highest overall performance.

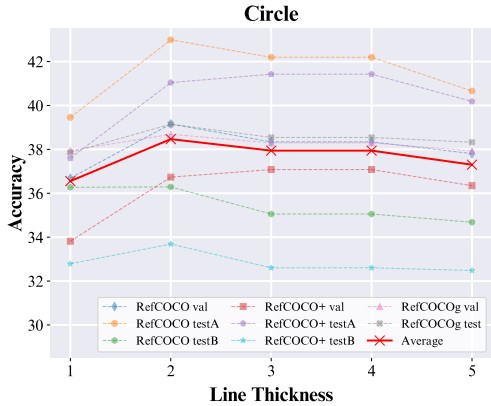

Figure S1: Ablation study on the line thickness.

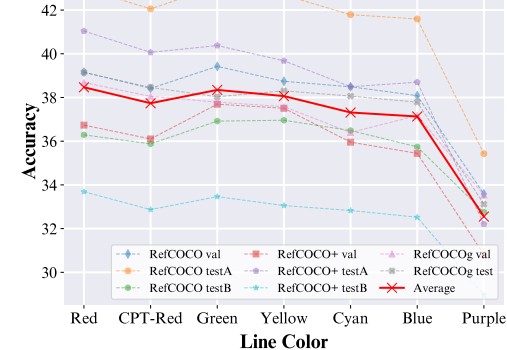

Figure S2: Ablation study on the line color.

---

*Corresponding authors.

37th Conference on Neural Information Processing Systems (NeurIPS 2023).

## A.2 Transparency and Color for Color-Based Prompting

For color-based prompting, such as in CPT [6], we consider two hyperparameters of the mask, namely transparency and color. Initially, we set the color to green and vary the transparency from 0.1 to 0.9 with a step of 0.1. Next, with a constant transparency of 0.5, we explore different candidate colors, which are used in the experiments of line-based prompts. Fig. S3 demonstrates that the transparency ranging from 0.3 to 0.5 achieves the best results. Since it effectively highlights the targets without causing excessive interference to the target information. Meanwhile, in Fig. S4, we observe that the impact of color on average accuracy is relatively minor compared to other hyperparameters. Surprisingly, the experiment reveals that the yellow mask achieves the best overall result.

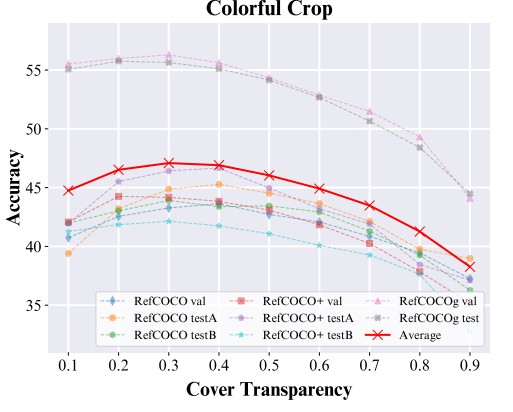

Figure S3: Ablation study on the cover transparency.     Figure S4: Ablation study on the cover color.

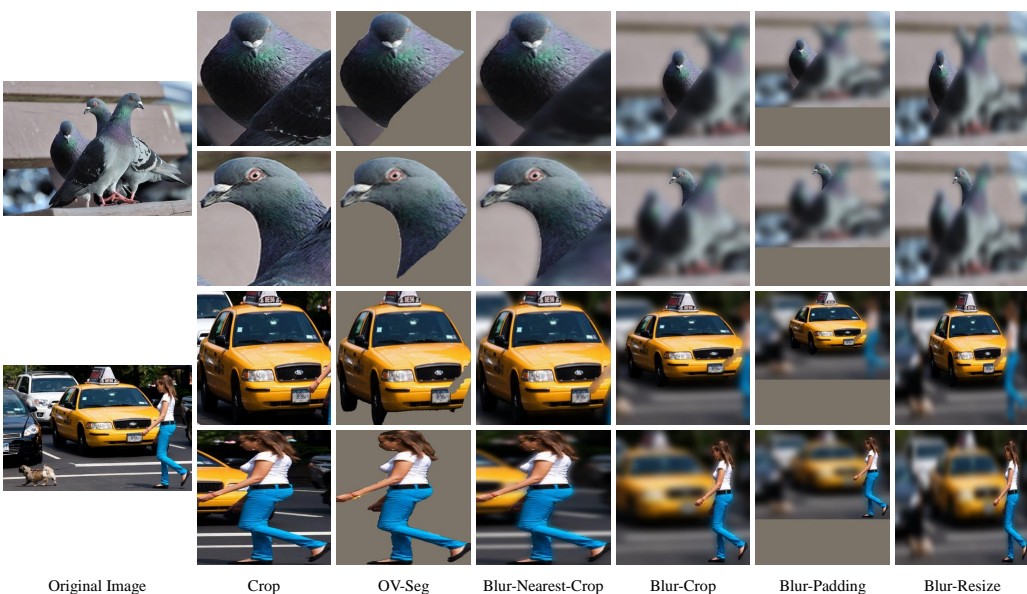

Figure S5: Visualization of different image preprocessing operations aimed at transforming the prompted image into a square format that is compatible with CLIP inputs.

## A.3 Image Preprocessing

Image preprocessing ((Fig. S5)) is a crucial but often overlooked aspect in visual prompting techniques used alongside VLMs, such as CLIP [3], for image-text alignment prediction. Typically, CLIP expects input images to be square, whereas most of the natural images in datasets are rectangular. Consequently, these images require preprocessing before being fed into the model. Previous studies [5, 4, 6]

have primarily resized rectangular images into squares. However, this approach may result in object stretching and deformation. As a result, it may cause the semantic information to deviate from its original meaning. Additionally, OV-Seg [2] investigate the open-vocabulary semantic segmentation by first generating cropped instances and subsequently applying grayscale masking to eliminate the background. This work introduces mask engineering, which aims to address the performance bottleneck resulting from masked images featuring a gray background. Then, it finetunes the CLIP to adapt for masked images, so that the accuracy and effectiveness of open-vocabulary semantic segmentation can be improved. However, we have empirically demonstrated that it may be unnecessary to introduce the incompatible background.

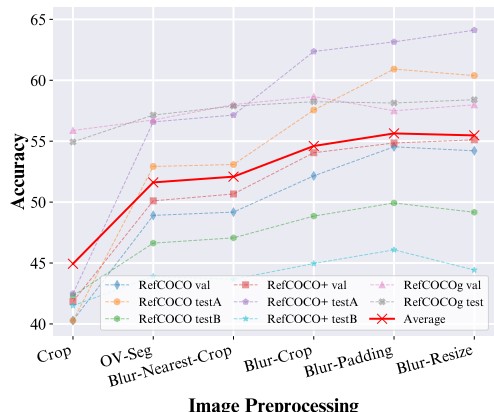

Figure S6: Ablation study on the image preprocessing.

By applying a single blur operation, we can retain more spatial relevance information. Moreover, since the images are blurred, they may have a relatively minor impact on the recognition ability of CLIP on the target. Further, some instances may not experience severe deformation compared to previous preprocessing operations, as illustrated by the pedestrian in Fig. S5. Notably, the "Bokeh" paradigm subtly corresponds to the web-scaled data used in CLIP training, where CLIP may possess embedded knowledge for recognizing blurred images. As shown in Fig. S6, blur-based prompting achieves overall better performance compared to crop-based and grayscale-based OV-Seg prompting. It is worth noting that equipping Blur Reverse Mask with further padding or resizing achieves the best performance.

## A.4 Robustness of Mask Preciseness

Given relatively accurate masks or corresponding circles, we investigate to analyze the impact of different mask preciseness on zero-shot performance. To explore this, we introduce a parameter called the "expand scale", which allows us to adjust the size of the mask around the target by expanding or shrinking it. When the expand scale is less than 1, it indicates shrinking towards the center. The visualization of this process is depicted in Fig. S7. Regardless of whether the mask is expanded or shrunk, it leads to inaccurate captioning of the target, resulting in a decrease in performance (Fig. S7). This emphasizes the significance of having a precise and detailed mask. To conduct a control experiment, we also applied the same operation to the Blur Reverse Circle. Similarly, the best performance is achieved when the scale is set to 1. An important reason is that a larger range introduces more background noise, while a smaller range leads to a loss of target information, both of which lead to a decrease in performance, as illustrated in Fig. S8.

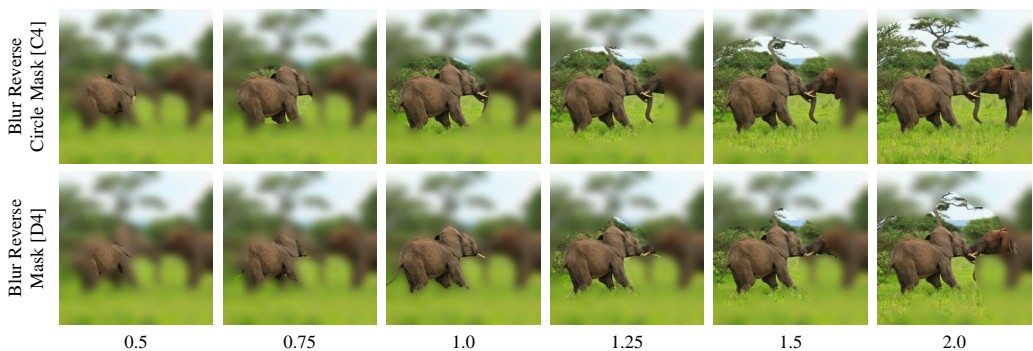

Figure S7: Visualization of blurring images with masks of different preciseness adjusted by an expand scale coefficient, denoted in the bottom. The "1.0" is the expand scale of the baseline for each visual prompting.

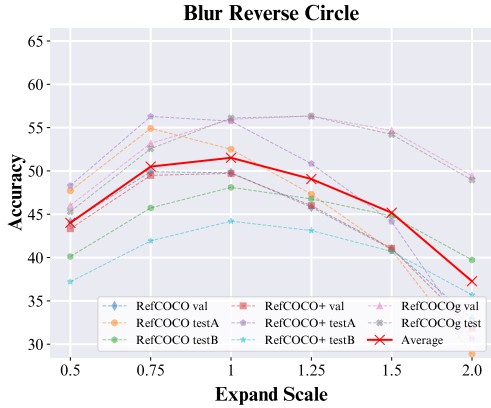
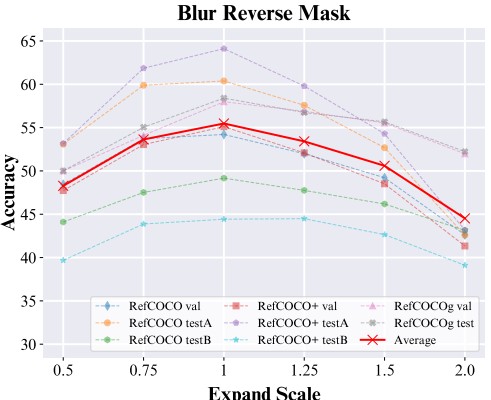

Figure S8: Ablation study on the robustness of the mask preciseness under Blur Reverse Circle.

Figure S9: Ablation study on the robustness of the mask preciseness under Blur Reverse Mask.

## B  Visual Prompting Expertise

In terms of general object detection tasks, cropping may be preferable since labels are frequently only relevant to the target image. As a result, it is critical to minimize background noise as much as possible. In the case of part detection and referring expression comprehension task, positive prompting or negative prompting with the preservation of more global information is more appropriate. This is because local features, such as a dog's tail, are difficult recognize in isolation. Regarding the referring expression comprehension task, it is similar to the caption and may contain information about the target object's relative relationship with other objects. When it is difficult to identify the target object, information about its relative position can be utilized for zero-shot localization.

## C  More Visualizations

In this section, we present the visualizations of the Fine-Grained Visual Prompting (FGVP) results for the referring expression comprehension task, as shown in Fig. S10. The captions are displayed in the lower-left corner of the displayed images. Additionally, image grounding results on the COCO dataset using input images and their possible corresponding labels are shown. (Fig. S11). The results reflect the alignment between instances and labels within the figure. Categories with the same labels are indicated using identical colors, and all candidates are generated from the Segment Anything method [1] under the framework we implemented.

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

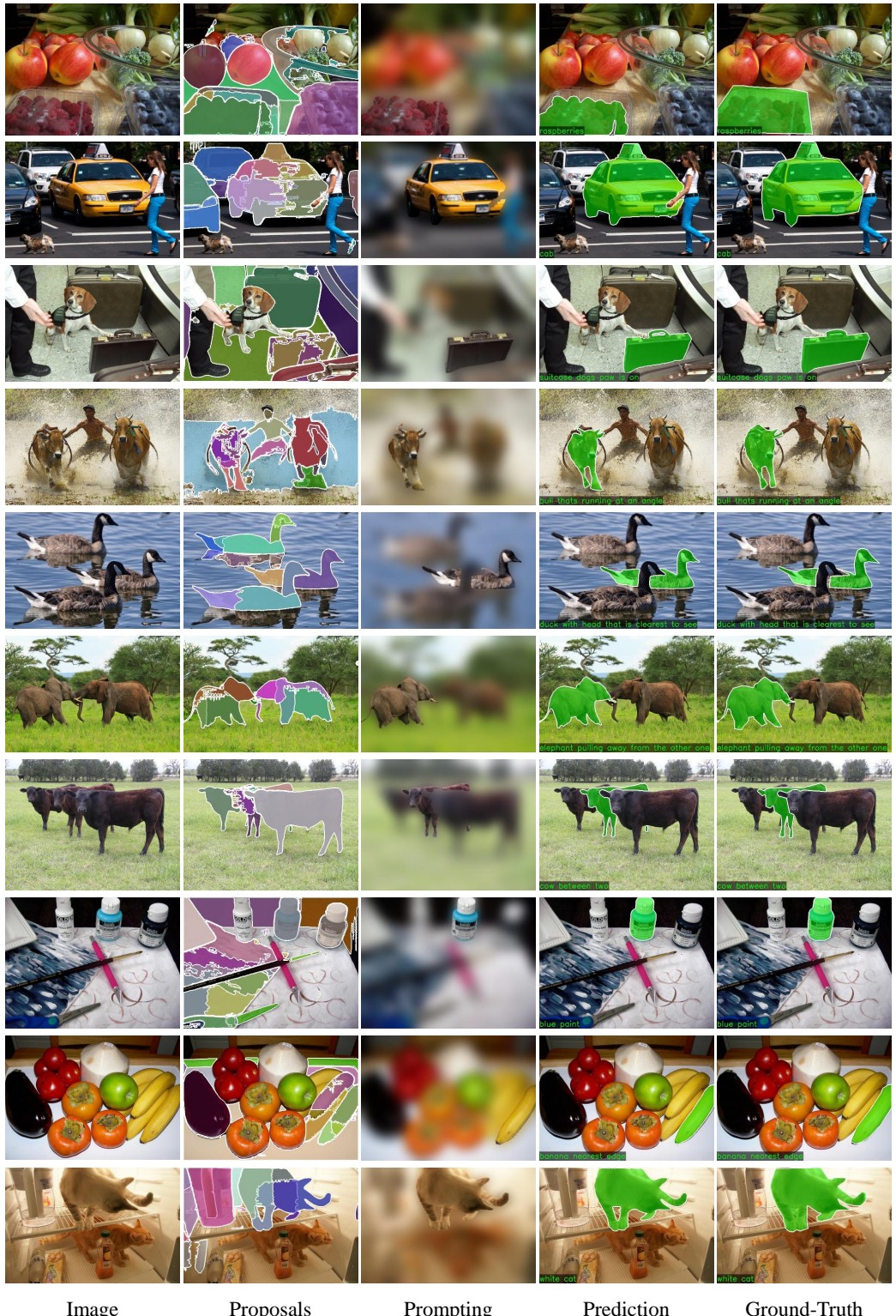

| Image | Proposals | Prompting | Prediction | Ground-Truth |

Figure S10: Visualization of FGVP results on the referring expression comprehension task under the RefCOCO, RefCOCO+, and RefCOCOg datasets.

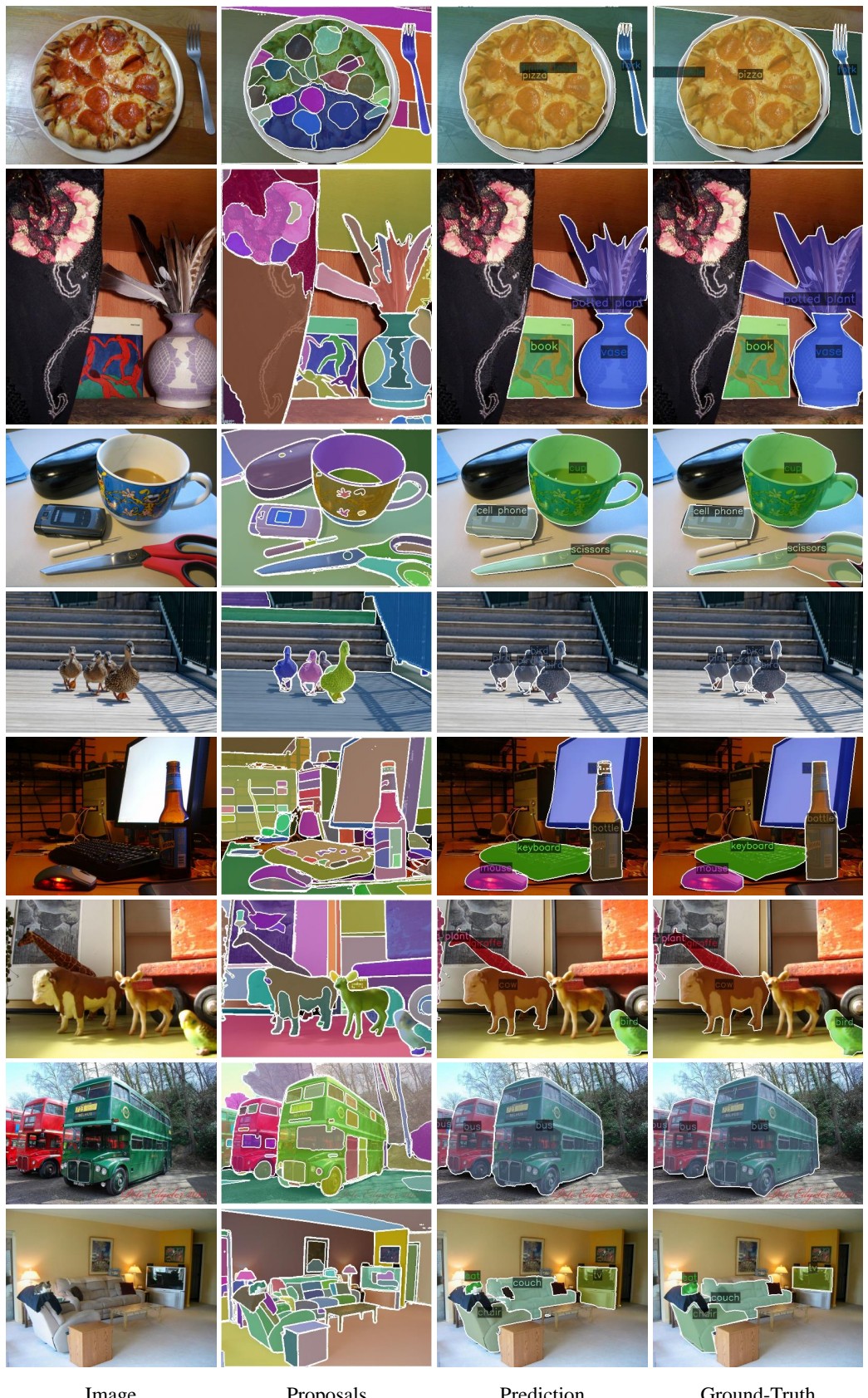

|       |          |            |              |
|-------|----------|------------|--------------|
| Image | Proposals | Prediction | Ground-Truth |

Figure S11: Visualization of FGVP results under the COCO dataset.