# OpenReview forum: "Fine-Grained Visual Prompting"
_NeurIPS.cc/2023/Conference — NeurIPS 2023 poster_

### Official Review · Reviewer_TfGG · 2023-06-30

**Soundness:** 3 good
**Presentation:** 3 good
**Contribution:** 3 good
**Rating:** 7
**Confidence:** 4

**Summary:**

This paper proposes Fine-Grained Visual Prompting (FGVP) that incorporates Blur Reverse Mask to improve the semantic localization capability of VLMs, like CLIP. It provides a comparison to other possible methods for highlighting the different parts/objects in the image, based on SAM and other techniques. The resulting method improves the performance on RefCOCO* datasets.

**Strengths:**

The paper is written in a clear way and describes the algorithm well. The evaluation shows clear improvements and the ablation study is convincing. The idea to blur and mask the background of different object/part proposals is both innovative and significant.

**Weaknesses:**

The related work section missing two relevant works in the visual prompting domain:

1. Bhang et al., "Exploring Visual Prompts for Adapting Large-Scale Models", 2022

2. Bar et al., "Visual Prompting via Image Inpainting", NeurIPS, 2022


Moreover, while SAM is a powerful method for segmentation, it requires running the model with a relatively dense grid of keypoints. This runtime can be significant and should be discussed in the limitations section.

**Questions:**

Can SAM be replaced with a simpler segmentation model, like color-based unsupervised segmentation methods?
Any ablation of SAM will be useful here for understanding the scale/quality that is required from the segmentation network for achieving good results.

**Limitations:**

The limitation section is present and addresses some unexplored directions. I suggest adding runtime estimates.

---

> ### Author Rebuttal · Authors · 2023-08-09
>
> Thank you for the comments and suggestions!
>
> **Q1**: Missing related works in the visual prompting domain.\
> **A1**: Thanks, we will cite these works in the revised version.
>
>
>
> **Q2**: The runtime should be discussed since the method requires running the model with a relatively dense grid of keypoints.\
> **A2**: Thanks for your advice, we will add and discuss the inference runtime limitations in the revised version.\
> **Firstly**, we propose a method that uses the proposal boxes from a detector to generate masks. Consequently, proposals are sparse and their quantity depends on the outputs of the detector. In this situation, the time comparison results are presented in our **global response A1**.\
> **Secondly**, under the framework where no detector is available and we need to use dense grid points as proposals, the runtime experiments are implemented in the **global response A2**.
>
>
>
> **Q3**: Can SAM be replaced with a simpler segmentation model, like color-based unsupervised segmentation methods? More ablation studies of SAM for understanding the scale/quality to achieve good results.\
> **A3**: **First of all**, SAM could be replaced with another segmentor; we implement the unsupervised mask generator in FreeSOLO to produce masks. Also, we use SAM of different scales to understand how the scale/quality affects the final results. The results can be found in the **response to Reviewer Xx88 A1**. \
> **Notably**, we also conducted more experiments in the supplementary materials (B.4 Robustness of Mask Precision) on the Robustness of Mask Precision by manually expanding or shrinking the mask derived from SAM. I hope it will provide more information about the mask quality.

---

### Official Review · Reviewer_Rf2R · 2023-07-03

**Soundness:** 3 good
**Presentation:** 3 good
**Contribution:** 2 fair
**Rating:** 7
**Confidence:** 4

**Summary:**

This paper proposes a new “visual prompting” method. Visual prompting refers to the idea of altering images to guide the “attention” of a vision-language model when the model is used to embed the image. For example, to obtain an embedding for an object in an image that contains many objects, the user could draw a red circle around the object of interest and then embed the image. Since the vision-language model might have seen images during training in which important objects are highlighted with red circles, the resulting embedding might focus on the circled object.

This approach can be used to partially solve tasks such as referring expression comprehension: Given a set of bounding boxes and text descriptions, visual prompting can be used to find the best-fitting description for each box. A separate object detector is needed to obtain the bounding boxes first.

The paper proposes a visual prompting method that consists of blurring everything in the image except for the object of interest. This is done by first using a pretrained segmentation model (Segment Anything) to obtain a mask for the object of interest, and then blur everything outside of the mask. The motivation for this approach is that it simulates the shallow depth of field seen in photographs taken with a large aperture, which are commonly found in VLM training data.

The blur method is compared to the “red circle” method and several variants in object detection and referring expression tasks. The blur method consistently performs best. In addition, hyperparameter sweeps for the blur radius and other hyperparameters are shown.

**Strengths:**

1. The proposed visual prompting method consistently improves over other methods.

2. The blurring approach is well motivated by the abundance of photographs with shallow depth of field. This is a clever way to exploit “natural supervision” present in large-scale web image training data.


**Weaknesses:**

1. The proposed method is significantly more complex than the “red circle” method, since it relies on a large segmentation model. The inference cost of the proposed method is therefore much higher than the “red circle” method. This should be acknowledged in the discussion and/or limitation sections.

2. While the proposed method works well, it is an incremental improvement over the “red circle” idea (https://arxiv.org/pdf/2304.06712.pdf) and the evaluation is not as comprehensive as in the “red circle” paper. For example, no analysis of the relative biases of red circle vs blur methods are performed, and no failure cases are discussed. It is not clear if the contribution is substantial enough for NeurIPS.

**Questions:**

Please use descriptive method names (keypoint/box/circle/…) instead of just A/B/C in Table2 and elsewhere, so that the reader doesn't have to refer to Figure 1 all the time.

**Limitations:**

The runtime cost of the proposed method needs to discussed further.

The biases of the method compared to the "red circle" method need to be evaluated and discussed.

A generic "broader impact" statement is given in the appendix.

---

> ### Author Rebuttal · Authors · 2023-08-09
>
> Thank you for the comments and suggestions!
>
> **Q1**: The inference cost of the proposed method.\
> **A1**: We present the inference cost comparison experiments in the **Global Response A1 and A2**. Thank you for your advice; we will acknowledge it in our limitation sections in the revised version.
>
>
> **Q2**: Analysis of the relative biases of red circle vs blur methods.\
> **A2**: Following the analysis section in the paper of RedCircle, we **first** select one figure in COCO containing a male and a female. We visualize their categories classified by CLIP with added criminal-related texts under different visual prompting methods. The comparison of relative biases among RedCircle and FGVP is depicted in **Figure R2 in the PDF in the global response**. From the figure, RedCircle tends to classify persons into criminal categories, while FGVP and the original image without prompting manage to classify correctly.
> It is mainly because a red circle is not a natural marking compared to the training web-scale data of CLIP. However, FGVP possesses more non-post-processed characteristics which helps to reduce the biases.
>
> **Next**, we quantify the biases following RedCircle based on the same datasets of FairFace and COCO. Additionally, we experimented with each visual prompting strategy on COCO with and without cropping the person out of the entire image as a pre-processing operation. From the following table, we can observe that although our FGVP might introduce a few more biases than the raw image, it substantially reduces biases compared to RedCircle due to its more natural prompting design.
>
>
> |Model|Visual Prompt|FairFace|COCO w/ crop|COCO w/o crop|
> |:-:|:-:|:-:|:-:|:-:|
> |ViT-L/14@336px|Crop|13.0|40.8|43.6|
> |ViT-L/14@336px|RedCircle|20.6 (+7.6)|49.9 (+36.9)|69.3 (+56.3)|
> |ViT-L/14@336px|FGVP|15.9 (+2.9)|34.1 (-6.7)|47.8 (+4.2)|
> |ViT-B/32|Crop|14.5|27.2|34.9|
> |ViT-B/32|RedCircle|22.0 (+7.5)|44.1 (+29.6)|68.6 (+54.1)|
> |ViT-B/32|FGVP|8.2 (-6.3)|19.5 (-7.7)|15.8 (-19.1)|
> |RN50×16|Crop|19.5|55.1|50.7|
> |RN50×16|RedCircle|38.5 (+19.0)|71.4 (+51.9)|72.6 (+53.1)|
> |RN50×16|FGVP|21.6 (+2.1)|56.0 (+0.9)|28.9 (-21.8)|
>
>
> **Q3**: Failure cases visualization and analysis. \
> **A3**: Please refer to the failure case visualizations in **Figures R4 and R5 in the PDF document** and the corresponding analysis in the **global response A3**. **Lastly**, it's essential to highlight that we've established a comprehensive framework to facilitate evaluation comparisons among diverse visual prompting techniques and their post-processing ensembles. These aspects were not addressed by RedCircle.
>
>
>
> **Q4**: Use descriptive method names (keypoint/box/circle/…) instead of just A/B/C in the Table.\
> **A4**: Thanks for pointing out this inconvenience. We will address this in the revised version.

---

> > ### Comment · Reviewer_Rf2R · 2023-08-15
> >
> > Thank you for your response. The response address my main questions. The additional analyses, in particular the bias analyses, provide a strong argument for using the proposed method over the "red circle" method. Also, the inference cost analysis shows that while the proposed method is more expensive, there are ways to speed it up and the additional cost is not excessive. I therefore raised my recommendation to "Accept".

---

> > > ### Author Response · Authors · 2023-08-22
> > > **Response to Reviewer Rf2R**
> > >
> > > We greatly appreciate your valuable feedback, as well as the time and dedication you invested in thoroughly reading and comprehending both our paper and our response.

---

### Official Review · Reviewer_Z75J · 2023-07-07

**Soundness:** 3 good
**Presentation:** 3 good
**Contribution:** 2 fair
**Rating:** 6
**Confidence:** 2

**Summary:**

This paper works on visual prompting. They proposed FGVP, together with Blur Reverse Mask, to improve the semantic localization ability of the vision-language model.

**Strengths:**

Solid experiments showing the effectiveness of their method.

**Weaknesses:**

In general, this is a good work. However, there’re several things concerning me:

- The novelty of this work is not well established. Seems like an engineering combination of previous works.
- The discussion of the upper bound is based on the assumption, while the legitimacy of the assumption is not well discussed.

**Questions:**

No further questions.

**Limitations:**

No notable limitations.

---

> ### Author Rebuttal · Authors · 2023-08-09
>
> Thank you for the comments and suggestions!
>
> **Q1**: The novelty of this work is not well established. Seems like an engineering combination of previous works.
> **A1**: **Firstly**, thank you for your concern. With the development of large vision-language models and segmentors, they potentially embed knowledge for various downstream tasks. However, their basic usage mainly focuses on global-level tasks and class-agnostic segmentation. **They cannot be easily adopted to achieve high performance in instance-wise localization and classification tasks.** Besides, additional training for specific tasks end-to-end would be costly. Therefore, developing a zero-shot architecture is an elegant solution, which can unleash the potential of large models and combine prior knowledge to effectively tackle certain tasks without specific tuning. Simultaneously, such an architecture can also serve as a good baseline for further improvement, providing deeper insights for related work.
> **Then**, we would like to reclaim our contributions and inspirations:
> - To better align target regions with their captions, we propose to **highlight the target instances according to their semantic mask by blurring the background as fine-grained visual prompting**.
> - After extensive experiments across diverse visual prompts, we demonstrate the superior performance of FGVP, which achieves SOTA performance on **zero-shot benchmarks**.
> - The background blurring strategy is aligned with the natural photography images in web-scale data used for training VLMs, which **exploits potential knowledge within large models**.
>
> **In conclusion**, as mentioned by the reviewer Xx88, our work “**presents a novel idea that using SAM to generate semantic masks as better visual prompts, which is an original idea not explored in prior works**” and “**is a clever way to exploit 'natural supervision' present in large-scale web image training data**” by the reviewer Rf2R. For these reasons, we believe it is promising to further explore and improve the FGVP.
>
>
>
> **Q2**: The discussion of the upper bound is based on the assumption, while the legitimacy of the assumption is not well discussed.
> **A2**: From the table in the response to **Reviewer Xx88 A1**, we conducted an ablation study of the performance under different mask qualities. The **ground truth masks achieve almost the best overall performance** compared to those generated by SAM or other segmentors. \
> **Notably**, the masks generated by SAM achieve competitive or even better results compared to GT. An important reason is that the GT mask in the COCO dataset is annotated with relatively rough polygons, which has a certain gap with the high-quality mask from SAM.\
> **In fact**, instead of considering the ground truth mask as an upper bound, it is mainly **served as a unified and constant evaluation benchmark to measure and compare different visual prompts**, as GT masks have already existed for convenient inference.

---

> > ### Comment · Reviewer_Z75J · 2023-08-16
> >
> > Thanks for your response. The response addresses most of my concerns. I thereby change my rating to 6.

---

> > > ### Author Response · Authors · 2023-08-22
> > > **Response to Reviewer Z75J**
> > >
> > > We greatly appreciate your valuable feedback, as well as the time and dedication you invested in thoroughly reading and comprehending both our paper and our response.

---

### Official Review · Reviewer_jMVP · 2023-07-10

**Soundness:** 3 good
**Presentation:** 3 good
**Contribution:** 3 good
**Rating:** 7
**Confidence:** 4

**Summary:**

This paper proposes a visual prompting method that exploits the segmentation masks of interested objects in images to generate more fine-grained visual prompts. Experiments show that the proposed methods achieve competitive results on zero-shot referring expressions comprehension and part detection.

**Strengths:**

1. This paper is well-organized. The motivation and the framework is clearly presented.
2. The experiments confirm the effectiveness of the visual prompt design.

**Weaknesses:**

1. The ablation studies of different VLMs are missing. Since visual prompting is a zero-shot framework, it is natural that the performance of the visual prompts differ on VLMs trained on different data. The paper adopts the CLIP as the VLM in all experiments. How do different visual prompts perform on other VLMs?
2. Figures 2 & 3 are very similar and thus redundant. The authors should consider merging them into one figure.

**Questions:**

1. Basically, I like the section 3.3 of the discussion of the prior knowledge introduced by the large amount of photography images in the large-scale image-text dataset. Could the authors provide more discussions on the analysis of the training data of the VLMs, and the effect of better alignments between the visual prompt design and the VLM training data?
2. How good are the zero-shot results compared with the few-shot / many-shot methods? The few-shot / many-shot methods may be provided for general reference.

**Limitations:**

The proposed visual prompts can be further verified on more object-based tasks.

---

> ### Author Rebuttal · Authors · 2023-08-09
>
> Thank you for the comments and suggestions!
>
> **Q1**: Ablation studies on different VLMs. \
> **A1**: The results across various VLMs demonstrate the consistent improvement of FGVP. Importantly, we observe that RedCircle experiences a significant performance decline when transitioning from CLIP to other models like SLIP, aligning with the results reported in the original RedCircle paper. In contrast, FGVP maintains its performance gain across different architectures.
>
> |Method|Backbone|Data|Params|Input size|Crop|RedCircle|FGVP|
> |:-:|:-:|:-:|:-:|:-:|:-:|:-:|:-:|
> |OpenAI CLIP|ViT-L/14@336px|CLIP-400M|304M|336|45.3|48.9|52.8|
> |OpenAI CLIP|ViT-B-32|CLIP-400M|87M|224|47.8|44.0|51.2|
> |Open CLIP|ViT-L/14|LAION-2B|304M|224|49.9|48.1|49.8|
> |SLIP|ViT-L/16|YFCC-15M|303M|224|44.3|33.7|49.3|
> |BLIP-v2|ViT-L/14|Merged-129M|304M|224|46.9|37.7|51.0|
> |EVA-02-CLIP|ViT-L/14@336px|Merged-2B|304M|336|51.4|51.5|55.8|
>
> **Q2**: Figures 2 & 3 are very similar and thus redundant. The authors should consider merging them into one figure.\
> **A2**: Thank you for your proposal. There are significant design differences between the two mentioned frameworks. One requires a detector, while the other does not. The masks in the first framework are sparse, tied to proposal boxes, whereas the latter generates dense masks from SAM prompted with grid points. Describing and showcasing the two frameworks separately will help highlight the differences in design and effect details. After careful consideration, we chose to keep both frameworks independent.
>
> **Q3**: More discussions on the training data of the VLMs, and the effect of better alignments between the data and visual prompt design.\
> **A3**: Thanks for valuing our discussion. We hope the biases analysis could provide more insight into the visual prompting-training data alignment. In **Figure R2 in PDF in the global response**, we added criminal text to regular COCO images with people. The RedCircle tends to classify the person into criminal classes while FGVP does not. We assume that a red circle is unnatural and lacks realism compared to web images. On the contrary, the image with a blurred background tends to look more natural and common rather than artificially post-processed, which helps to reduce unexpected bias. For detailed quantitative results, please refer to **Response to Reviewer Rf2R A2**.
>
> **In an ideal scenario**, our prompt may be better aligned with VLM training data by factoring in image depth. This involves gradual depth-based blurring incorporated with depth estimation results. Capturing natural shallow depth-of-field could be better than the current uniform blur.
>
> **Q4**: Compared with the few-shot / many-shot methods.\
> **A4**: We present the best performance as reported in the original paper for the methods listed below. The results are partly summarized from the original papers of Pseudo-Q [1] and CPT [2].\
> [1] Jiang H, Lin Y, Han D, et al. Pseudo-q: Generating pseudo language queries for visual grounding. CVPR, 2022.\
> [2] Yao Y, Zhang A, Zhang Z, et al. Cpt: Colorful prompt tuning for pre-trained vision-language models. arXiv preprint arXiv:2109.11797, 2021.
>
>
> ||||RefCOCO|RefCOCO|RefCOCO|RefCOCO+|RefCOCO+|RefCOCO+|RefCOCOg|RefCOCOg|
> |:-:|:-:|:-:|:-:|:-:|:-:|:-:|:-:|:-:|:-:|:-:|
> |Method|Published|Supervision|val|test-A|test-B|val|test-A|test-B|val|test|
> |MAttNet|CVPR’18|Full|76.7|81.1|70.0|65.3|71.6|56.0|66.6|67.3|
> |NMTree|ICCV’19|Full|76.4|81.2|70.1|66.5|72.0|57.5|65.9|66.4|
> |FAOA|ICCV’19|Full|72.5|74.4|68.5|56.8|60.2|49.6|61.3|60.4|
> |ReSC|ECCV’20|Full|77.6|80.5|72.3|63.6|68.4|56.8|67.3|67.2|
> |TransVG|ICCV’21|Full|80.3|82.7|78.1|63.5|68.2|55.6|67.7|67.4|
> |VC|CVPR’18|Weak|\--|33.3|30.1|\--|34.6|31.6|\--|\--|
> |ARN|ICCV’19|Weak|34.3|36.4|33.1|34.5|36.0|33.8|\--|\--|
> |KPRN|ACMMM’19|Weak|35.0|34.7|37.0|36.0|35.2|37.0|\--|\--|
> |DTWREG|TPAMI’21|Weak|39.2|41.1|37.7|39.2|40.1|38.1|\--|\--|
> |CPT|ArXiv’21|8-shot|41.3|48.2|35.7|42.6|49.3|35.4|47.4|47.4|
> |CPT|ArXiv’21|4-shot|40.7|47.4|35.3|40.3|46.5|34.5|44.4|44.4|
> |CPT|ArXiv’21|2-shot|39.8|45.6|33.9|38.6|44.5|32.8|44.7|44.3|
> |CPT|ArXiv’21|1-shot|37.2|41.5|33.2|37.9|42.3|33.9|43.1|43.4|
> |CPT|ArXiv’21|zero-shot|32.2|36.1|30.3|31.9|35.2|28.8|36.7|36.5|
> |Pseudo-Q|CVPR’22|zero-shot|56.0|58.3|54.1|38.9|45.1|32.1|46.3|47.4|
> |ReClip|ArXiv’22|zero-shot|45.8|46.1|47.1|47.9|50.1|45.1|59.3|59.0|
> |RedCircle|ArXiv’23|zero-shot|49.8|58.6|39.9|55.3|63.9|45.4|59.4|58.9|
> |FGVP (ours)|ArXiv’23|zero-shot|59.6|65.0|52.0|60.0|66.8|49.7|63.3|63.4|
>
>
> **Q5**: The proposed visual prompts can be further verified on more object-based tasks.\
> **A5**: Thank you for your advice. We plan to extend FGVP to more grounding and open vocabulary benchmarks in our future works. Here, we provide experiments on two more object-based tasks.\
> **1)** **Firstly**, we extended FGVP to open vocabulary detection based on the current state-of-the-art work OV-Seg [1].\
> [1] Liang F, Wu B, Dai X, et al. Open-vocabulary semantic segmentation with mask-adapted clip. CVPR, 2023.
> |Method|Segmentor backbone|Clip|PAS-20|
> |:-:|:-:|:-:|:-:|
> |OVSeg|Swin-B|ViT-L|94.5|
> |OVSeg-Blur|Swin-B|ViT-L|95.1|
>
> **2)** **Next**, we experimented with FGVP on the Referring Image Segmentation benchmark. The table shows that we outperformed the current state-of-the-art Global-Local CLIP [2] by replacing feature cropping with our FGVP.\
> [2] Yu S, Seo P H, Son J. Zero-shot Referring Image Segmentation with Global-Local Context Features. CVPR, 2023.
>
> |Method|Visual Encoder|oIoU|mIoU|
> |:-:|:-:|:-:|:-:|
> |Cropping|ViT-B/32|22.7|24.8|
> |Global-Local CLIP|ViT-B/32|24.8|26.2|
> |FGVP+Global-Local CLIP|ViT-B/32|25.1|26.6|

---

> > ### Comment · Reviewer_jMVP · 2023-08-21
> > **Thanks for the rebuttal**
> >
> > Thank the authors for their response. Since the rebuttal addresses most of my concerns, I raise my score to "accept".

---

> > > ### Author Response · Authors · 2023-08-22
> > > **Response to Reviewer jMVP**
> > >
> > > We greatly appreciate your valuable feedback, as well as the time and dedication you invested in thoroughly reading and comprehending both our paper and our response.

---

### Official Review · Reviewer_Xx88 · 2023-07-14

**Soundness:** 3 good
**Presentation:** 3 good
**Contribution:** 2 fair
**Rating:** 6
**Confidence:** 4

**Summary:**

The paper proposes Fine-Grained Visual Prompting (FGVP), which uses precise semantic masks from SAM as visual prompts to improve spatial localization of vision-language models (VLMs) like CLIP for instance-level tasks. The key contributions are:

- Systematically study different visual prompting techniques like cropping, boxes, circles, masks, etc. Show that blurring background outside target mask (Blur Reverse Mask) works best.
- Achieve state-of-the-art results on referring expression comprehension benchmarks RefCOCO/RefCOCO+/RefCOCOg, outperforming prior works.
- Demonstrate FGVP can enable zero-shot part detection on PACO dataset without needing any box proposals, again outperforming other prompting methods.

**Strengths:**

Originality: The paper presents a novel idea that using SAM to generate semantic masks from detected bbox as better visual prompts of specific instance in the image, which is an original idea not explored in prior works.

Quality: The overall approach is technically sound. The experiments follow standard protocols and are extensive in studying different visual prompting design. The results demonstrate benefits over existing methods.

Clarity: The paper is well-written and easy to follow. The problem context, proposed method, experiments are clearly explained. Figures and tables aid understanding.

Significance: FGVP pushes state-of-the-art in two important vision-language tasks - referring expression and part detection. The analysis may inspires more research on the properties of VLMs regarding spatial understanding.

**Weaknesses:**

While the paper presents a novel fine-grained visual prompting technique and achieves state-of-the-art results, there are some aspects where the analysis could be strengthened:

More Insights from Study: The paper performs an extensive set of experiments on different visual prompt designs. However, it could provide a more detailed analysis of the inferences and insights derived from this study. For instance, comparing the gap between using ground truth and predicted bboxs would give insights into the impact of mask quality. Explaining the differences in various design choices (VP, PP, proposals etc.) in Table 3 would be informative. Attention visualizations could help reveal why fine-grained prompting is more beneficial.

Computational Overhead: The paper proposes generating semantic masks using SAM models. However, the computational overhead this introduces is not analyzed. Reporting inference times, scalability, etc. would provide a better understanding of its practical viability.

Joint Language and Visual Prompting: The study is currently limited to exploring only visual prompts. Evaluating prompts on both visual and textual modalities could offer a more comprehensive understanding of VLMs. It is good that authors discussed this point in the limitation though.

Granular and Failure Analysis: Providing per-class performance breakdowns and detailed failure analysis compared to other methods through examples would provide useful insights into where the improvements come from.

In summary, while the core ideas are promising, performing a more thorough empirical analysis along the above dimensions would strengthen the paper and provide a better understanding of the factors behind the efficacy of fine-grained visual prompting.

**Questions:**

See the main suggestions in weakness.

- For the RegCOCO/RefCOCO+/RefCOCOg results in Table 4, which subsets are the numbers reported on?
- Line 288-290 if there are better grid size / NMS thresholds, how do we still use the suboptimal default ones?

**Limitations:**

Limitations are discussed in the paper

---

> ### Author Rebuttal · Authors · 2023-08-09
>
> Thank you for the comments and suggestions!
>
> **Q1**: More insights from the study.\
> **A1**: \
> **1)** Compare the gap in mask quality caused by using ground truth and predicted boxes.\
> **Firstly**, Table 2 in the main paper contrasts results with masks derived from ground truth (left side) and proposal boxes (right side).\
> **Secondly**, we broadened experiments by employing SAM at varied scales and an unsupervised mask generator from FreeSOLO, yielding diverse mask qualities. We present the performance of FGVP with the Blur Reverse Mask prompting. Generally, FGVP achieves the highest accuracy with ground truth masks, reinforcing the rationale for "Using Ground Truth as Performance Upper Bound." Larger SAM backbones typically yield better masks, resulting in higher performance, while the unsupervised mask generator results in the lowest performance for poor mask qualities.
>
> |Mask source|SAM scale|COCO|PACO|RefCOCO|RefCOCO+|RefCOCOg|
> |:-:|:-:|:-:|:-:|:-:|:-:|:-:|
> |GT mask|\--|67.8|42.7|52.8|58.0|63.5|
> |GT box|huge|68.0|39.5|52.3|56.1|62.2|
> |GT box|large|67.9|39.8|51.6|55.6|61.3|
> |GT box|base|67.4|39.1|52.1|55.2|60.8|
> |UNINEXT proposal|huge|\--|\--|52.8|55.4|57.8|
> |UNINEXT proposal|large|\--|\--|52.2|54.8|57.5|
> |UNINEXT proposal|base|\--|\--|52.1|54.5|58.8|
> |Mask generator of FreeSOLO|\--|17.4|11.6|27.7|30.5|38.2|
>
>
> **2)** Design choices with the visual prompt (VP), post-processing (PP), proposals, etc. in Table 3.\
> **Above all**, the guiding principle for all settings is to maintain consistency with the compared works.
> **To be specific**:
> - **CPT and our codebase** focus on individual VP performance without PP. We use UNINEXT and MAttNet proposal banks to demonstrate the robustness of our enhancements. It's important to note that different proposal selections solely affect the box candidates, which are equitably shared among all the comparison prompting methods.
> - **For the ReClip codebase**, the ReClip employs cropping and colorful boxes as visual prompts, with default spatially-relations post-processing. To ensure a fair comparison, we **first** add cropping as an ensembled VP to all experiments. **Next**, to facilitate comparison with RedCircle (which inherently uses Score Subtraction as post-processing and primarily ensembles based on three circle VPs, as summarized in Table 1), we adopt the same three types of prompt formats but based on semantic masks. **Finally**, we aim to explore a higher performance possible under various VP and PP ensembles.
>
>
> **3)** Attention visualizations between fine-grained prompting and previous methods.\
> Following the pipeline established in DINO, we present attention visualizations in **Figure R3 of the PDF in the global response**. We compare Crop, RedCircle, and FGVP using the RefCOCO datasets. The green number in the top left corner denotes the similarity score and is arranged in descending order. Each correct prediction is indicated by a red rectangle.\
> The visualization results show the reasons for the superior performance of FGVP and the limitations of RedCircle. This is because FGVP demonstrates a more reasonable behavior in terms of reducing attention to weakly correlated or distracting backgrounds and enhancing focus on the target object, especially small objects. While RedCircle can also alter attention allocation, the degree is quite limited, resulting in inferior performance. \
> At the same time, the attention analysis effectively validates the main perspective of our paper regarding the capability of the reverse blur mask to reduce attention on weakly correlated pixels and enhance focus on the main subject. Therefore, the proposed FGVP holds better potential in the field.
>
> **Q2**: Experiments on the computational overhead, inference times, and scalability.\
> **A2**: Please refer to the **global response A1 and A2** for detailed information.
>
> **Q3**: Joint Language and Visual Prompting.\
> **A3**: We appreciate your insightful comment and constructive guidance and have included two experiments for better understanding.\
> **1)** **Firstly**, there is a general approach to text prompts that employs SPACY to extract nouns from captions. For instance, given a caption like "a dog on the table," the noun extracted would be "dog." Subsequently, the similarity score $S$ can be derived through an ensemble of the caption score $S_c$ and the prompted noun score $S_n$ : $S = r \times S_c + (1 - r) \times S_n$, where $r$ represents the balance ratio.
>
> ||caption ($r$=1)|noun ($r$=0)|ensemble ($r$=0.5)|
> |:-:|:-:|:-:|:-:|
> |example|a dog on the table|dog|\--|
> |FGVP|52.8|53.1|54.4|
>
> **2)** **Furthermore**, we have explored strategies for rephrasing text based on visual prompts. For instance, when presented with an image prompt featuring a blurred background, the original text "a dog on the table" could be rephrased as "a dog on the table with a blurred background".
>
> ||caption|<caption> with blurred background|
> |:-:|:-:|:-:|
> |example|a dog on the table|a dog on the table **with blurred background**|
> |FGVP|52.8|54.8|
>
> **Q4**: Per-class performance and comprehensive failure analysis compared to other methods.\
> **A4**: Please refer to the **global response A3**.
>
> **Q5**: Other issues.\
> **A5**: \
> **1)** The performance reported in Table 4 is derived from the validation set of all datasets.\
> **2)** Using the suboptimal settings in Table 4 while there are better grid size / NMS Thresholds.\
> The grid size and NMS threshold we employed are default settings used in the official codebase of SAM. We avoided a higher setting due to increased inference cost and lengthy times, as discussed in **global response A2**. \
> **Regarding your query**, an ablation study with larger settings (NMS threshold = 0.9, grid size = 32) is included, showing the continued superiority of our FGVP.
>
> ||PACO|RefCOCO|RefCOCO+|RefCOCOg|
> |:-:|:-:|:-:|:-:|:-:|
> |Crop|19.5|17.4|22.1|35.3|
> |RedCircle|19.9|25.9|31.0|35.0|
> |FGVP|23.2|42.1|46.0|50.7|

---

> > ### Comment · Reviewer_Xx88 · 2023-08-21
> > **Reviewer comment after rebuttal**
> >
> > I really appreciate the author's rebuttal. Most of my concerns are addressed. Therefore, I raise my score by one. However, I do think the paper could go a bit deeper when it comes to the insights from the empirical studies on which prompting works best.
> >
> > Take for example the "middle zebra" and "second guy from right white shirt" mentioned in the rebuttal attachment. These seem to hinge on understanding spatial relationships. Perhaps they'd benefit more from a global context than just crop-based prompting. But with blurring, it seems we're still cutting down on that global context info, making things trickier.
> >
> > Also, I'm curious about the per-class performance analysis - why does FGVP fall short compared to RedCircle for things like the baseball glove and traffic light? Digging deeper into these questions and sharing insights would really help in understanding FGVP better. It would be great to get a clearer picture of what makes one visual prompting strategy stand out over another.

---

> > > ### Author Response · Authors · 2023-08-22
> > > **Response to Reviewer Xx88**
> > >
> > > We greatly appreciate your valuable feedback, as well as the time and dedication you invested in thoroughly reading and comprehending both our paper and our response. Here is some more analysis regarding your question.
> > >
> > > ### 1) Understanding of spatial relationships.
> > > **Essentially**, the referring captions can be separated into two parts: **relative position description** and **object nouns**. Changing from cropping to blurring can preserve the relative positional relationships while also reducing background noise. However, it does indeed decrease the global contextual information, leading to less confidence in terms of "spatial position" compared to the vanilla image. **Yet**, for the confidence in "object nouns," I think both RedCircle and the original image would fall behind, as the model would need to handle more focal attention on other objects.
> > >
> > > Taking the example of "the second guy from the right wearing a white shirt," without blurring, the model would also have to consider the confidence of "object nouns" across multiple background objects, thereby affecting the scoring on the actual target. **Therefore**, I think blurring **achieves a good trade-off in terms of confidence in both relative position description and object nouns**.
> > >
> > > ### 2) Per-class performance analysis.
> > > **Empirically**, blurring doesn't perform well when identifying small target objects.
> > > **To find out**, we further computed the average proportion between the target size and the total image size for each category. The results reveal that RedCircle outperforms blur in categories where the instances occupy only 3% of the image, while categories where blur is superior occupy around 10% of the image size.
> > > **Moreover**, from **Figure R5 in the PDF of the global response**, we can observe that although both methods utilize fine-grained masks as the foundation for prompting, with blur, the emphasis is on weakening the background to highlight the target, while the other approach involves applying colorful masks to the target area for positive marking. This visualization indicates that **for small target objects, the positive mask-type annotations tend to yield better results**. **Notably, RedCircle is also a positive marking strategy**. Categories such as "baseball glove" and "traffic light," which, on average, only occupy 0.9% and 0.7% of the entire image, respectively, fall into the category of extremely small objects.
> > > **In fact**, when we employ positive masks as visual prompts, we achieve superior results compared to RedCircle for these two categories. This particular design concept for visual prompts is summarized in Section 3.2 of the paper.

---

### Author Rebuttal · Authors · 2023-08-09

Thank you to all the reviewers for their valuable comments and suggestions! Here are the responses to some common concerns.

**Q1**: Experiments of inference cost with **detector proposals (Figure 2)**.\
**A1**: We conducted efficiency experiments, comparing inference costs in terms of computation and speed between our method and others. As FGVP relies on SAM for semantic masks, we ablate the **scalability** with various SAM backbone scales. Notably, the post-processing technique to filter small disconnected regions and holes in masks can further improve performance at the cost of speed. Disabling the mask-filter post-processing will **greatly improve the speed without losing too much performance**. Experiments are run on RefCOCO with a CLIP pretrained ViT-L/14@336px on 8×NVIDIA A100. Generally, FGVP takes more inference times than others, which will be acknowledged as a limitation and improvement direction in the revised version.

|Visual Prompt|SAM scale|Mask-filter|CUDA memory (GB)|Inference time (min)|Image per GPU second|Acc|
|:-:|:-:|:-:|:-:|:-:|:-:|:-:|
|Crop|\--|\--|0.91|4.49|5.03|45.3|
|RedCircle|\--|\--|0.91|4.00|5.64|48.9|
|FGVP|base|no|1.32|5.20|4.34|51.7|
|FGVP|base|yes|1.32|27.47|0.82|52.1|
|FGVP|large|no|2.14|6.29|3.59|51.0|
|FGVP|large|yes|2.14|27.49|0.82|52.2|
|FGVP|huge|no|3.42|7.34|3.08|51.9|
|FGVP|huge|yes|3.42|28.02|0.81|52.8|

**Q2**: Experiments of inference cost with **dense grid points as proposals (Figure 3)**.\
**A2**: Without provided detectors and proposals, we utilize SAM with dense grid points and an NMS threshold for mask filtering. We explore speed-performance trade-offs by varying grid sizes and NMS thresholds. All visual prompting methods, whether boxes, circles, or masks, rely on SAM for proposal yielding. Thus, the NMS threshold and grid size affect all methods equally. Experiments on PACO with a CLIP pretrained ViT-L/14@336px and SAM-huge on 8×NVIDIA A100 show that **FGVP could outperform RedCircle in speed and accuracy** at grid size 8 and NMS threshold 0.95 trade-off.

|Visual Prompt|Grid size|NMS threshold|Inference time (min)|Image per GPU second|Acc|
|:-:|:-:|:-:|:-:|:-:|:-:|
|Crop|16|0.7|13.34|3.27|16.5|
|Crop|32|0.95|37.25|1.17|19.5|
|RedCircle|16|0.7|12.75|3.42|17.4|
|RedCircle|32|0.95|34.18|1.28|19.9|
|FGVP|8|0.7|8.33|5.24|17.3|
|FGVP|8|0.95|9.17|4.76|20.5|
|FGVP|16|0.7|14.89|2.93|18.4|
|FGVP|16|0.95|17.29|2.52|22.0|
|FGVP|32|0.7|34.73|1.26|19.0|
|FGVP|32|0.95|39.66|1.10|23.2|

**Q3**: Granular and Failure Analysis\
**A3**: \
**1)** Per-class performance.\
We mainly compare the per-class performance of RedCircle and our FGVP on COCO. FGVP (Blur Reverse Mask) surpasses RedCircle in class accuracy for 53 out of 80 classes, particularly excelling in major categories like car, person, bird, and chair. However, FGVP is inferior to RedCircle in a few categories like the baseball glove and traffic light. We show all results in **Figure R1 in PDF in the global response**.

|Category|person|car|bird|chair|traffic light|tennis racket|
|:-:|:-:|:-:|:-:|:-:|:-:|:-:|
|Instance Number|10777|1918|427|1771|634|225|
|RedCircle|35.6|23.6|58.8|32.5|67.4|87.1|
|FGVP|76.2 (+40.6)|62.5 (+38.9)|91.6 (+32.8)|62.7 (+30.2)|52.8 (-14.6)|45.3 (-41.8)|

**2)** Detailed failure analysis compared with other methods.\
We visualize the challenging cases in **RefCOCO (Figure R4) and COCO (Figure R5) in the PDF file in the global response**.
- **For the RefCOCO dataset**, we find that all visual prompts performed poorly when the target object is difficult to be recognized due to its semantic gap or weak perceptual difference from the background. For example, in the sample Figure R4 (1), the "white shirt" mentioned in the caption is hardly visible in the image. In sample (2), FGVP grounds part of the "left bicycle" that is not cut off/blocked to the front wheel of the actual bicycle. At the same time, it is considered that the rest of the actual bicycle may belong to another bicycle. This is largely due to perceptual illusion. Samples (3) and (4) typically indicate that the performance bottleneck of FGVP could lie within SAM, yielding inaccurate masks and masks containing unrelated noises.
- **For the COCO dataset**, we observed that small objects can be accurately localized using positive masks rather than blurred masks. This indicates an improvement over the specific usage of semantic masks as FGVP for different types of instances.

**Consequently**, these difficult samples pose a general challenge to the algorithm. Overcoming such challenges becomes an important avenue for our future research. For instance, enhancing the involvement and connection of textual cues and exploring strategies like gathering and tuning visually difficult samples could be explored.

---

### Decision · Program_Chairs · 2023-09-21

**Decision:**

Accept (poster)

**Comment:**

This paper has received overwhelmingly positive reviews with reviewers praising the originality, significance and clarity of the proposed approach in tackling the vision-language tasks of referring expression and part detection.
In addition to these strengths, reviewers also appreciated the strong empirical results on the mentioned tasks that demonstrate state-of-the-art performance on multiple datasets.
Finally, the rebuttals provided by the authors included additional experiments, ablation studies and observations that reviewers indicated as satisfactory in addressing some of the remaining questions raised in the initial reviews.